# Control of leucine-dependent mTORC1 pathway through chemical intervention of leucyl-tRNA synthetase and RagD interaction

Jong Hyun Kim[1], Chulho Lee[2,3], Minji Lee[2], Haipeng Wang[4], Kibum Kim[2], Seung Joon Park[2], Ina Yoon[1], Jayun Jang[1], Hanchao Zhao[5], Hoi Kyoung Kim[1], Nam Hoon Kwon[1], Seung Jae Jeong[1], Hee Chan Yoo[6], Jae Hyun Kim[2,3], Jee Sun Yang[3], Myeong Youl Lee[7], Chang Woo Lee[7], Jieun Yun[7], Soo Jin Oh[7], Jong Soon Kang[7], Susan A. Martinis[5], Kwang Yeon Hwang[8], Min Guo[4], Gyoonhee Han[2,3], Jung Min Han[2,6] & Sunghoon Kim[1,9]

Leucyl-tRNA synthetase (LRS) is known to function as leucine sensor in the mammalian target of rapamycin complex 1 (mTORC1) pathway. However, the pathophysiological significance of its activity is not well understood. Here, we demonstrate that the leucine sensor function for mTORC1 activation of LRS can be decoupled from its catalytic activity. We identified compounds that inhibit the leucine-dependent mTORC1 pathway by specifically inhibiting the GTPase activating function of LRS, while not affecting the catalytic activity. For further analysis, we selected one compound, BC-LI-0186, which binds to the RagD interacting site of LRS, thereby inhibiting lysosomal localization of LRS and mTORC1 activity. It also effectively suppressed the activity of cancer-associated *MTOR* mutants and the growth of rapamycin-resistant cancer cells. These findings suggest new strategies for controlling tumor growth that avoid the resistance to existing mTOR inhibitors resulting from cancer-associated *MTOR* mutations.

[1] Medicinal Bioconvergence Research Center, College of Pharmacy, Seoul National University, Gwanak-gu, Seoul 08826, South Korea. [2] Department of Integrated OMICS for Biomedical Science, Yonsei University, Seodaemun-gu, Seoul 03722, South Korea. [3] Translational Research Center for Protein Function Control, Department of Biotechnology, Yonsei University, Seodaemun-gu, Seoul 03722, South Korea. [4] Department of Cancer Biology, The Scripps Research Institute, Scripps Florida, Jupiter, FL 33458, USA. [5] Department of Biochemistry, University of Illinois at Urbana, Urbana, IL 61820, USA. [6] College of Pharmacy, Yonsei University, Seodaemun-gu, Seoul 03722, South Korea. [7] Bioevaluation Center, Korea Research Institute of Bioscience and Biotechnology, Ochang, Chungbuk 363-883, South Korea. [8] Division of Biotechnology, College of Life Sciences and Biotechnology, Korea University, Seongbuk-gu, Seoul 136-701, South Korea. [9] Department of Molecular Medicine and Biopharmaceutical Sciences, Graduate School of Convergence Science and Technology, Seoul National University, Gwanak-gu, Seoul 08826, South Korea. Jong Hyun Kim and Chulho Lee contributed equally to this work. Correspondence and requests for materials should be addressed to J.M.H. (email: jhan74@yonsei.ac.kr) or to S.K. (email: sungkim@snu.ac.kr)

Amino acids not only serve as substrates for protein synthesis but also control protein metabolism[1]. Sensing of intracellular amino acid availability is mediated by mammalian target of rapamycin complex 1 (mTORC1), which controls many cellular processes such as protein synthesis, autophagy, and cell growth, and is implicated in human diseases including cancer, obesity, diabetes, and neurodegeneration[2–5]. Thus, understanding of amino acid signaling to mTORC1 is crucial for developing strategies to control relevant pathophysiology. Mammals express four Rag GTPases—RagA, B, C, and D[6], which are the central mediators in this pathway. Rag GTPases form obligate heterodimers of either RagA/C or RagB/D that mediate amino acid-induced mTORC1 activation[7–9]. Amino acids induce translocation of mTORC1 to lysosome, where the Rag heterodimers containing GTP-bound RagB serve as a docking site for mTORC1[10]. Leucine and glutamine stimulate mTORC1 by Rag GTPase-dependent and Rag GTPase-independent mechanisms[11].

Aminoacyl-tRNA synthetases are essential enzymes not only required for protein synthesis but also involved in diverse cellular physiological responses. In addition to their canonical role in ligating amino acids to their cognate tRNAs[12, 13], they also appear to control protein homeostasis by sensing amino acid availability. For instance, leucyl-tRNA synthetase (LRS) functions as a leucine sensor for mTORC1 by its activity as a GTPase-activating protein (GAP) for RagD[14]. Cdc60, a yeast form of LRS, interacts with Rag GTPase Gtr1 of the yeast EGO complex in a leucine-dependent manner and mediates leucine signaling to TORC1[15].

Many hyperactive and drug-resistance mTOR mutations have been identified in human cancers[16–19]. For example, everolimus, an allosteric inhibitor of mTOR, is effective in treating tumors with alterations in mTOR signaling. However, tumors have acquired resistance to everolimus due to mTOR mutations that block its ability to bind to the drug[19]. Another drug resistance mutation that confers resistance to rapamycin occurs in a conserved serine residue, S2035, in mTOR, which is crucial for the binding of FKBP12-rapamycin[20–22]. Thus, new therapeutic strategies are needed to overcome the resistance to current mTOR inhibitors. Here, we have identified compounds that specifically block the leucine-sensing function of LRS by interfering with its interaction with RagD, without affecting its catalytic activity. The selected compound BC-LI-0186 efficiently inhibited leucine-dependent mTORC1 activity and the growth of cancer cells that express drug-resistant *MTOR* mutations.

## Results

### Identification of leucine signaling inhibitor via LRS. Since LRS can influence protein synthesis via its activity in the mTORC1 pathway or tRNA charging, we investigated whether the two activities could be decoupled. For this, we first sorted 167 compounds from 5000 chemicals based on their structural similarities to leucinol, the leucine analog[23], and tested them for their ability to inhibit leucine-dependent S6K phosphorylation (> 90% at 100 μM) (Fig. 1a, b). The screening selected 12 compounds that were then used as the structural basis for further synthesis of 174 additional pyrazolone derivatives. The second screening (> 70% inhibition at 20 μM) identified 21 hits (Fig. 1c). Comparing their efficacy on mTORC1 activity, cell growth and death, solubility and predicted pharmacological behavior[24] (Supplementary Table 1), we finally selected BC-LI-0186 for further studies (Fig. 1d).

BC-LI-0186 bound to LRS (Fig. 1e) with a KD of 42.1 nM as determined by surface plasmon resonance (SPR) analysis, and inhibited leucine-dependent and isoleucine-dependent S6K T389 phosphorylation (Supplementary Fig. 1a) with an $IC_{50}$ of

81.4 nM (Fig. 1f, g), while not affecting AKT S473 phosphorylation (Fig. 1f), or glutamine-dependent or arginine-dependent S6K phosphorylation (Supplementary Fig. 1a). It showed no effect on the activities of 12 different kinases (Supplementary Fig. 1b), or the catalytic activities of cytosolic and mitochondrial LRS as measured by leucylation of tRNA$^{Leu}$ (Supplementary Fig. 1c). The compound did not inhibit the amino acid activation activity of cytosolic LRS as measured by leucyl-AMP formation even at 100 μM (Supplementary Fig. 1d). BC-LI-0186 did have a slight effect on the ability of LRS to edit misacylated tRNA$^{Leu}$ in the range of 10–100 μM (Supplementary Fig. 1e); however, this effect did not appear to be responsible for the inhibition of mTORC1 activity, since AN2690, a known inhibitor of LRS editing activity[25], did not affect leucine-induced mTORC1 or GCN2 activities (Supplementary Fig. 1f). SPR analysis indicated that BC-LI-0186 does not bind to Sestrin2, another leucine-binding protein[26] (Supplementary Fig. 1g). These results suggest that BC-LI-0186 can inhibit mTORC1 activity by diminishing its activity to bind to LRS, but without affecting its catalytic activity.

### Determination of the compound binding site in LRS. We found that the solubility of BC-LI-0186 was not sufficient to study the binding mode to LRS and therefore synthesized a structural analog of BC-LI-0186 with higher solubility that we designated BC-LI-0198 (Supplementary Table 1 and Supplementary Fig. 2a). Like BC-LI-0186, this compound also bound directly to LRS (Supplementary Fig. 2b) and specifically inhibited leucine-dependent S6K phosphorylation (Supplementary Fig. 2c).

Human cytosolic LRS (aa 1–1176) comprises the catalytic (CD, aa 1–255 and 515–763), CP1/editing (CP1, aa 256–514), tRNA anticodon-binding (ABD, aa 764–892), VC (aa 893–1063), and C-terminal UNE-L (aa 1064–1176) domains[27] (Fig. 2a). We modeled human LRS structure from the crystal structure of tRNA$^{Leu}$-bound *Pyrococcus horikoshii* LRS (PDB 1WZ2B) (32% sequence identity and 89% coverage), using the protein fold recognition server Phyre2 (Fig. 2a). The modeled structure showed domain arrangement similar to those of other species[28–31] in which the CD and ABD domains are tightly packed and the CP1 and VC domains are linked to the two ends of CD and ABD domains, respectively (Fig. 2a). The Robetta program was used for ab initio structural model prediction for the C-terminal UNE-L domain. The five best models were built based on the crystal structures of the chain A of linear ubiquitin and antibody Fab fragment complex (PDB 3U30A), both of which share predicted folds with UNE-L; the model that best fitted to small-angle X-ray scattering (SAXS) envelopes was chosen for further studies (Fig. 2a). The extra SAXS densities beyond the *hs*LRS structure may be from lower resolution of the SAXS, and potentially, suggest the flexibility of VC domain in LRS structure.

To understand the mechanism of action of BC-LI-0198, we conducted SAXS and hydrogen/deuterium exchange (HDX)-MS assays for LRS bound to BC-LI-0198 and Leu-AMS, a leucyl-sulfamoyl-adenylate analog of leucyl adenylate. Comparison of the SAXS data between the naked and BC-LI-0198-bound LRS revealed that the overall shape was similar with no major module rearrangement. The BC-LI-0198-bound structure was more elongated and thin, indicating conformational changes throughout the LRS (Rg and $D_{max}$ were 47.02 ± 1.01 Å and 205 Å for the naked LRS, and 54.19 ± 2.74 Å and 230 Å for the BC-LI-0198-bound LRS) (Fig. 2b and Supplementary Fig. 3a). Interestingly, we found that upon binding to BC-LI-0198, LRS showed an increase in general HDX, consistent with the increase of SAXS envelope size. In contrast, only a few LRS regions showed decreased deuterium exchange. One region was mapped to the VC domain, with a boundary of aa 958–974 for LRS with BC-LI-0198, and a

slight large boundary of aa 956–974 for LRS with BC-LI-0198/Leu-AMS (Fig. 2c and Supplementary Fig. 3b). The other regions were in the CD and CP1 domains, and decreases in deuterium exchange were recovered by the addition of Leu-AMS (Supplementary Fig. 3c). BC-LI-0198 neither induced a melting temperature shift (Supplementary Fig. 3d), nor inverted the effect of Leu-AMS in thermal shift assays, suggesting the compound does not affect LRS stability.

In the modeled human LRS structure, aa 958–974 residues of the VC domain are expected to be externally facing, forming a hydrophilic cluster within the 1770.1 Å$^2$ area on the surface of the VC domain. We performed a docking study of BC-LI-0186 with the VC domain of LRS and found that the most frequent docking positions are consistent with the HDX data (Fig. 2d, e). To further

confirm the BC-LI-0186-binding region of LRS, we introduced a point mutation at S974 and used SPR to analyze how it affected BC-LI-0186 binding to LRS. BC-LI-0186 showed significantly decreased affinity to the S974A mutant (Fig. 2f), suggesting the importance of the VC domain of LRS for its ability to bind BC-LI-0186.

**Mechanism of action of BC-LI-0186.** The VC domain of LRS was previously suggested to interact with RagD[14]. To test this hypothesis, we performed alanine scanning mutations in this region, and found that alanine mutations at H958, E960, K965, D968, and K970 decreased the interaction of LRS with RagD (Fig. 3a), suggesting the potential involvement of these residues

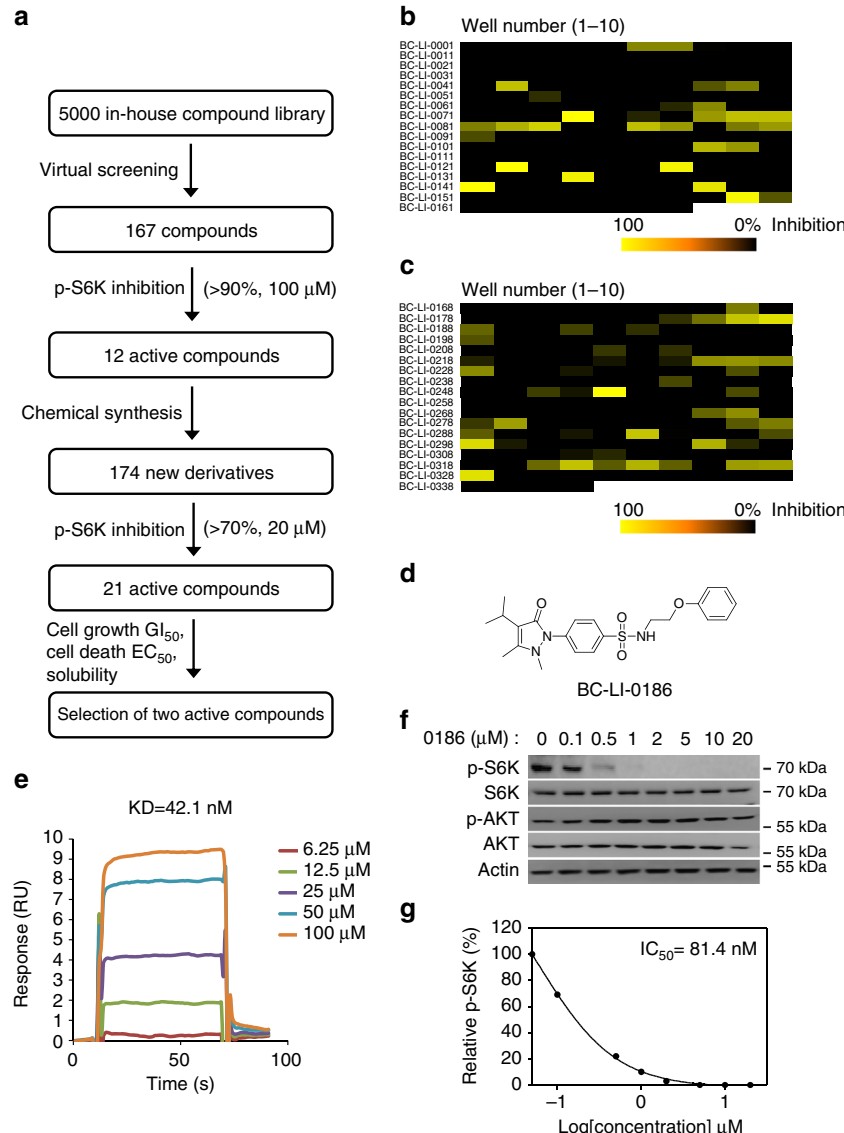

**Fig. 1** Identification of the compound inhibiting leucine-induced mTORC1 activity. **a** Schematic summary of the chemical screening for the mTORC1 inhibitor via LRS. **b** Level of leucine-induced S6K phosphorylation was monitored with 167 synthetic compounds. From the screening, 12 compounds that inhibited leucine-induced S6K phosphorylation more than 90% at 100 μM were selected. **c** Level of leucine-induced S6K phosphorylation was monitored with 174 additional synthetic compounds. From the screening, 21 compounds that inhibited leucine-induced S6K phosphorylation more than 70% at 20 μM were selected. Finally, two active compounds were selected based on their effects on mTORC1 activity, cell growth and death, as well as chemical solubility. **d** Chemical structure of BC-LI-0186. **e** The binding of BC-LI-0186 to LRS WT was determined by SPR as described in Methods. The *inset* represents the KD value between LRS WT and BC-LI-0186. **f** Effect of BC-LI-0186 on S6K phosphorylation was determined by Western blotting. AKT phosphorylation (S473) was monitored as a negative control. **g** Normalized band intensity of S6K phosphorylation in **f** was quantified and displayed as line graph. The *inset* represents the IC$_{50}$ value of BC-LI-0186

for the RagD binding. BC-LI-0186 inhibited LRS–RagD interaction with an $IC_{50}$ of 46.11 nM (Fig. 3b, c). Next, we examined the specificity of BC-LI-0186 for its interaction with LRS and RagD. BC-LI-0186 and its derivative BC-LI-0198 specifically inhibited the interaction of LRS and RagD, but had little effect on interactions of LRS-Vps34[32], LRS-EPRS[33], RagB-RagD pairs[7], and the core complex formation of mTORC1[34] (Fig. 3d). Using LRS mutants that retained their capacity to bind to WT RagD, we performed an in vitro pull-down assay to determine the extent to which their interaction with RagD was affected by the compound.

Interestingly, LRS mutations at R956, K957, and S974 were less affected by BC-LI-0186 than the wild-type and other mutants (Fig. 3e), suggesting that the residues located just next to the RagD-binding region are involved in the binding to the compound. Thus, it appears that there is overlap between the compound-binding and RagD-binding regions, although the residues involved may differ. We then examined whether the S974A mutant would render LRS-RagD resistant to the inhibitory effect of BC-LI-0186. Whereas both of the LRS–RagD interactions were significantly inhibited by BC-LI-0186 in LRS WT or

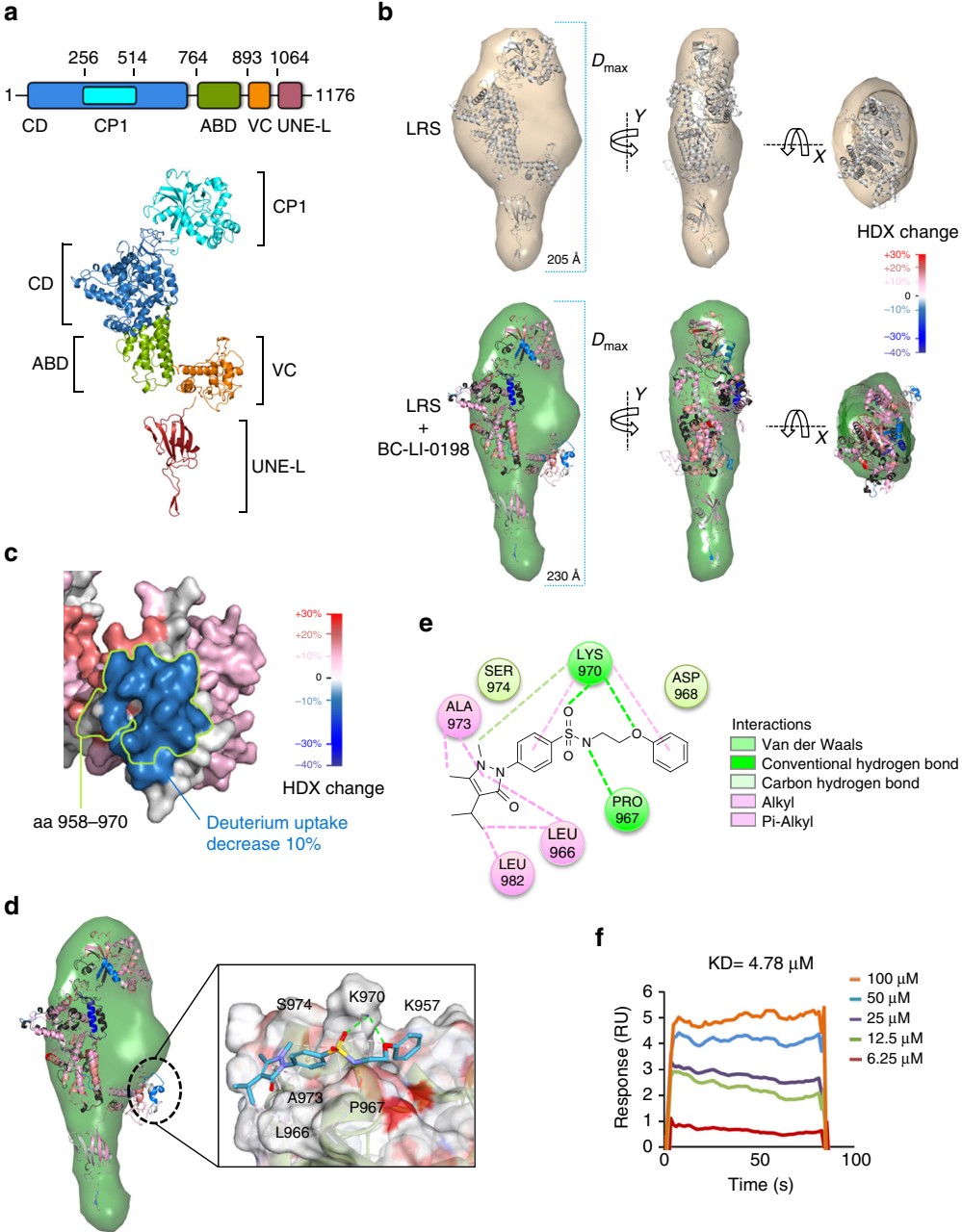

**Fig. 2** Determination of BC-LI-0186 docking site. **a** Structural modeling of human cytoplasmic LRS. *Upper*: Schematic representation for the domain arrangement of LRS. *Lower*: The structural model of human LRS, which fit to the LRS SAXS envelope. **b** SAXS and HDX-MS analysis of the BC-LI-0198 effect on LRS conformation. Structural models of the naked LRS (*upper*) and BC-LI-0198-bound LRS (*lower*). $D_{max}$ values of SAXS envelopes were indicated by *dashed lines*. *Colors* indicate changes of deuterium uptake compared with naked LRS as shown in the *color gradient bar*. **c** Surface representation of deuterium uptake changes of the LRS residues critical for the interaction with RagD upon BC-LI-0198 binding. Difference in deuterium uptake changes was shown in surface colors (*color gradient bar*). The residues critical for the interaction were circled by *lemon line*. **d** Structural modeling of BC-LI-0186 docking position in the pocket of LRS based on HDX-MS analysis. **e** The chemical binding residues are shown as *circles*. **f** The binding of BC-LI-0186 to LRS S974A mutant was determined by SPR as described in Methods. The *inset* represents the KD value between LRS S974A and BC-LI-0186

S953A-expressing cells, they became insensitive in the S974A mutant-expressing cells (Fig. 3f).

We then examined the effect of BC-LI-0186 on leucine-induced lysosomal localization of LRS by monitoring co-localization of LRS with the lysosomal marker LAMP2. The lysosomal localization of LRS was increased by the treatment of leucine[14] but not in the presence of BC-LI-0186, as shown by confocal microscopy (Supplementary Fig. 4a, b). The lysosomal

localization of LRS and Raptor, a component of mTORC1, was suppressed by BC-LI-0186 as well as BC-LI-0198 (Supplementary Fig. 4c). Together, these results suggest that BC-LI-0186 specifically inhibits the interaction between LRS and RagD.

**Validation of LRS as the specific target for BC-LI-0186.** Since LRS is known to serve as the GAP of the RagD GTPase[14], we

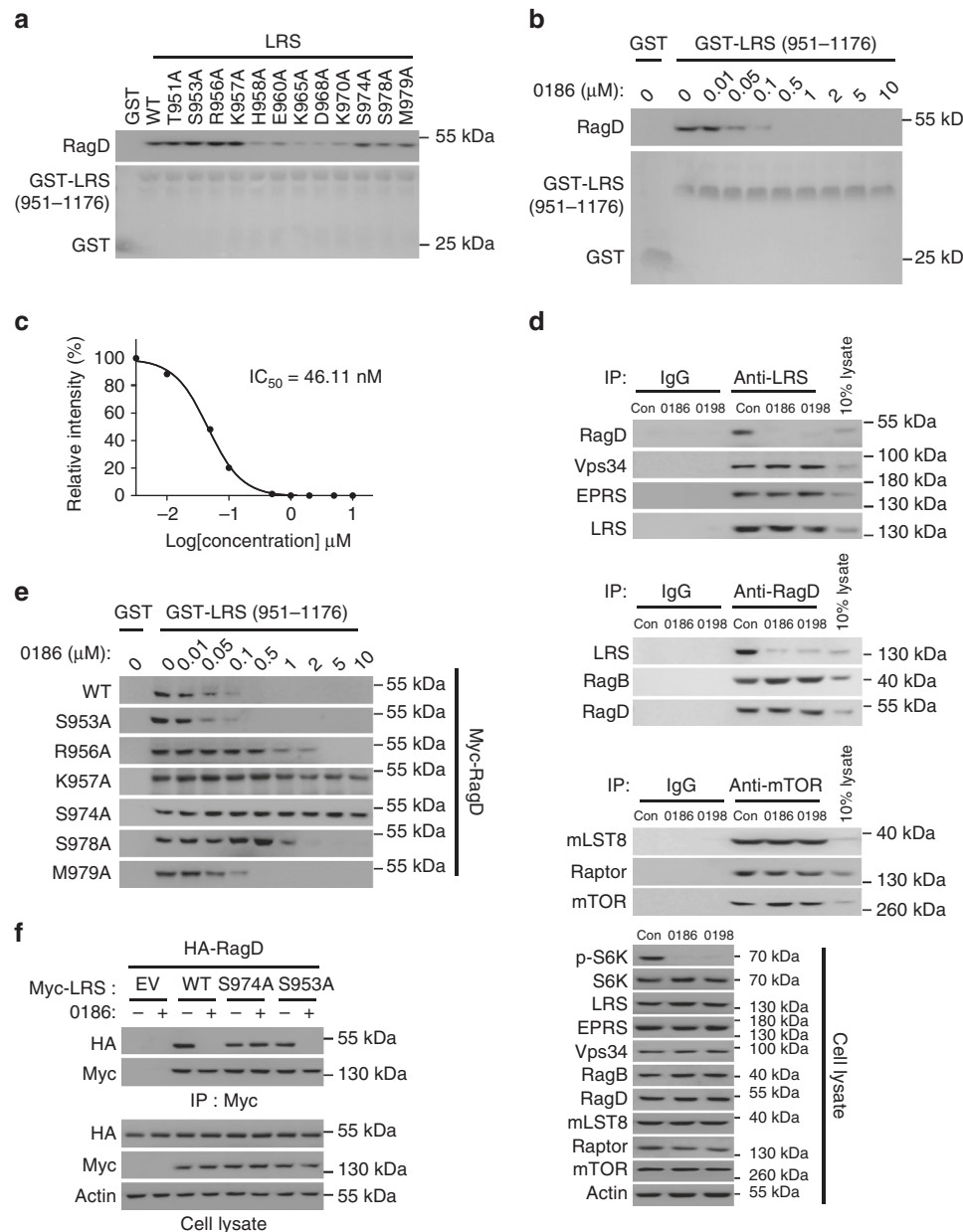

**Fig. 3** Chemical inhibition of the LRS–RagD interaction. **a** Effect of alanine mutations at the indicated residues located in the aa 951–981 peptide region of LRS VC domain on the interaction of LRS and RagD was determined by GST pull-down assays. Purified GST or GST-LRS (aa 951–1176) proteins were incubated with Myc-RagD^WT-transfected SW620 cell lysates, precipitated with glutathione sepharose beads and the precipitation of Myc-RagD was analyzed by immunoblotting with anti-Myc antibody. **b** Effect of BC-LI-0186 on the interaction of LRS WT with RagD was determined by in vitro pull-down of GST-LRS (957–1176 aa) and Myc-RagD. **c** Relative band intensity of Myc-RagD in **b** was quantified and displayed as *line graph*. The *inset* represents the IC$_{50}$ value of BC-LI-0186. **d** Effect of BC-LI-0186 and BC-LI-0198 on the endogenous interaction of LRS with RagD. Cells were treated with 10 μM BC-LI-0186 for 1 h and cell lysates were subjected to immunoprecipitation with anti-LRS, anti-RagD, or anti-mTOR antibodies. Co-immunoprecipitation was confirmed by immunoblotting with the indicated antibodies. **e** Effect of BC-LI-0186 on the interaction of LRS WT and the indicated mutants with Myc-RagD was determined by co-immunoprecipitation. **f** SW620 cells were co-transfected with Myc-tagged LRS WT, S974A, or S953A mutant and HA-tagged RagD WT. Cells were treated with 10 μM BC-LI-0186 for 1 h and cell lysates were subjected to immunoprecipitation with anti-Myc antibody. Cellular levels of the indicated proteins were analyzed by immunoblotting with their specific antibodies

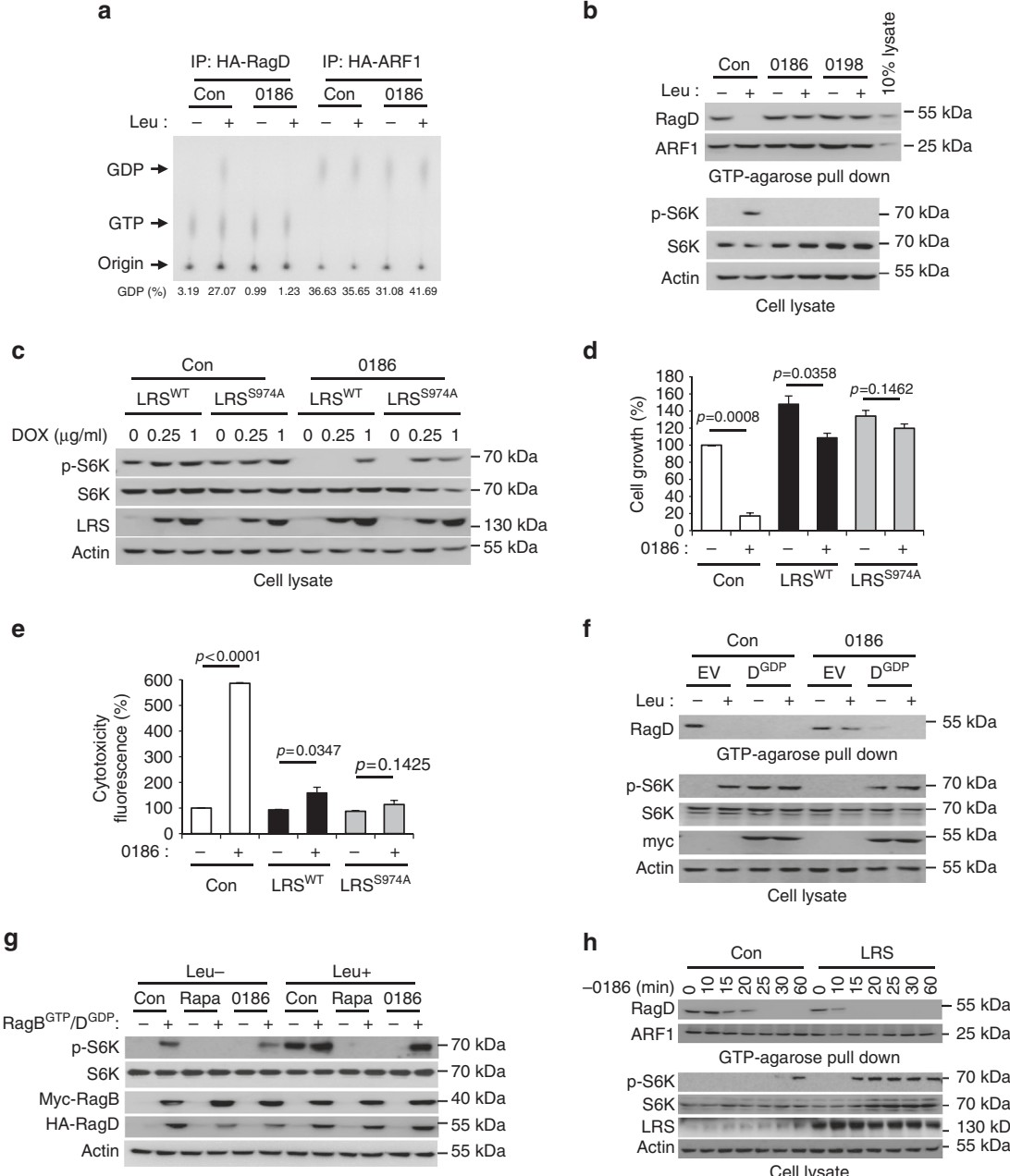

**Fig. 4** Chemical validation of LRS role in the control of RagD GTPase and mTORC1. **a** HA-RagD WT or HA-ARF1 was transfected into SW620 cells. After 24 h, the cells were incubated with 100 μCi/ml $^{32}$P-orthophosphate for 8 h, starved for leucine for 90 min, and then re-stimulated with leucine for 10 min. Cells were treated with 10 μM BC-LI-0186 during leucine starvation and re-stimulation. HA-RagD or HA-ARF1 was immunoprecipitated with anti-HA antibody and the bound nucleotides were eluted and analyzed by TLC. GDP% means GDP/(GDP + GTP) × 100. **b** Effect of BC-LI-0186 and BC-LI-0198 on the leucine-induced change of GTP hydrolysis of RagD. GTP-agarose bead pull-down assays were used to monitor the GTP-bound RagD or ARF1. After cells were treated with 10 μM BC-LI-0186 for 1 h, cell lysates were pulled down with GTP-agarose beads and the precipitated proteins with the beads were analyzed by immunoblotting with anti-RagD and anti-ARF1 antibodies. **c** Effect of LRS WT or S974A overexpression on S6K phosphorylation inhibited by BC-LI-0186. Inducible LRS WT or S974A-overexpressed SW620 cells were treated with 10 μM BC-LI-0186 for 1 h. Cell lysates were analyzed by immunoblotting with anti-p-S6K (T389), anti-S6K, anti-LRS, and anti-actin antibodies. **d** Effect of LRS WT or S974A overexpression on BC-LI-0186-induced growth inhibition (**d**) and cell death (**e**) (see Supplementary Fig. 5a). The data in Supplementary Fig. 5a were quantified and displayed as bar graphs. The *error bars* represent mean ± S.D. (*n* = 3). **f** Effect of RagD$^{GDP}$ on the BC-LI-0186-dependent inhibition of GTP hydrolysis of RagD and S6K phosphorylation. SW620 cells were transfected with GDP mutant of RagD (S77L) and then starved for leucine for 1 h and re-stimulated with leucine for 10 min in the presence or absence of 10 μM BC-LI-0186. Cell lysates were pulled down with GTP-agarose beads and the precipitated proteins with the beads were analyzed by immunoblotting with anti-RagD antibody. **g** Effect of RagB$^{GTP}$/RagD$^{GDP}$ on rapamycin or BC-LI-0186-dependent inhibition of S6K phosphorylation. SW620 cells were co-transfected with GDP mutant of RagD (S77L) and GTP mutant of RagB (Q99L) and then starved for leucine for 1 h and re-stimulated with leucine for 10 min in the presence or absence of 10 μM BC-LI-0186. **h** Effect of LRS overexpression on the kinetic changes of RagD and S6K phosphorylation that was arrested by BC-LI-0186 pretreatment. SW620 cells were treated with 10 μM BC-LI-0186 for 1 h. Then, cells were incubated with BC-LI-0186-free media for the indicated times. Cell lysates were pulled down with GTP-agarose beads and the precipitated proteins with the beads were analyzed by immunoblotting with anti-RagD or ARF1 antibodies (*upper*). Cell lysates were analyzed by immunoblotting with their specific antibodies (*lower*). The *error bars* represent mean ± S.D. (*n* = 3)

examined the effect of BC-LI-0186 on GTP hydrolysis of RagD by either monitoring $^{32}$P-labeled GTP-loading or GDP-loading of RagD, or by pull-down assay of RagD using GTP-agarose as previously described[35-38]. Consistent with the previous result[14], leucine stimulation of cells increased the GTP hydrolysis of RagD, but BC-LI-0186 suppressed leucine-induced GTP hydrolysis of RagD without any effect on GTP hydrolysis of ARF1 (Fig. 4a). In the in vitro pull-down assay, the GTP-binding form of small GTPases are pulled down with the GTP-agarose beads, and the GTP loaded form of RagD determined by Western blotting with a specific antibody. We found in the presence of leucine, the level of GTP-loaded RagD was decreased, whereas S6K phosphorylation was increased in SW620 cells. However, we did not observe leucine-induced GTP hydrolysis of RagD and S6K phosphorylation in the presence of BC-LI-0186 or BC-LI-0198 (Fig. 4b).

To further confirm that LRS is the specific target for BC-LI-0186, we compared the effect of LRS WT or S974A mutant on the activities of this compound in S6K phosphorylation, cell growth and death, using a DOX-inducible system. When we compared the extent of BC-LI-0186-dependent inhibition of S6K phosphorylation between the wild-type and S974A mutant-expressing cells, the mutant-expressing cells showed higher resistance to the compound than those expressing the wild-type LRS (Fig. 4c). Similarly, the S974A mutant-expressing cells showed higher resistance to the growth suppressive and cell death-inducing activities of BC-LI-0186 than the WT-expressing cells (Fig. 4d, e, respectively, see Supplementary Fig. 5a). BC-LI-0186-dependent autophagy induction (determined by the increase of LC3-II[39]) was also ablated by the induction of LRS (Supplementary Fig. 5b), further supporting LRS as the functional target of BC-LI-0186.

Ectopic introduction of RagD$^{GDP}$ or active heterodimer (RagB$^{GTP}$/RagD$^{GDP}$), both of which mimic the activated downstream effector of LRS, also compromised the effect of BC-LI-0186, but not rapamycin, on S6K phosphorylation inhibition (Fig. 4f, g), cancer cell growth inhibition, and cell death induction (Supplementary Fig. 5c–e). We then examined whether the BC-LI-0186-dependent arrest of the GTP to GDP conversion of RagD was relieved by increasing LRS expression. After depriving SW620 cells of the compound under LRS overexpression conditions, we monitored the conversion of the GTP to GDP form of RagD, and found that it was enhanced by overexpression of LRS (Fig. 4h). The LRS-dependent GTP hydrolysis of RagD also showed a positive correlation with S6K phosphorylation. All of these results validate LRS as the functional target in BC-LI-0186's inhibition of the mTORC1 pathway via RagD GTPase.

**Cellular effects of the LRS-binding compound**. We further examined the effect of BC-LI-0186 on cancer cell growth and death using human colon cancer SW620 cells. BC-LI-0186-induced cell death was determined by detecting Annexin V and propidium iodide (PI)-positive populations by flow cytometry (Fig. 5a). Using SW620 cells stably expressing nuclear-labeled red fluorescent protein (RFP), cell growth and death were monitored by counting the cells with red and green fluorescence (CellTox green), respectively. Within 24 h of BC-LI-0186 treatment, cell death was induced (detected with green color; see Experimental Procedures) without dramatically changing cell number (red color). However, 48 h after BC-LI-0186 treatment, cell number was also severely decreased (Fig. 5b). The BC-LI-0186 concentrations required for 50% growth inhibition ($GI_{50}$) and 50% cell death ($EC_{50}$) were determined as $11 \pm 0.97$ nM and $62 \pm 3.48$ nM, respectively (Fig. 5c). BC-LI-0186 treatment first inhibited mTORC1 activity, accompanied by PARP or procaspase-3 cleavage (Fig. 5d, e). We also observed that the caspase

inhibitor Q-VAD-FMK suppressed BC-LI-0186-dependent activation of caspase3/7 and cell death (Fig. 5f, g), indicating that BC-LI-0186 induced cell death through mTORC1 inhibition.

We then determined the effect of BC-LI-0186 on the proliferation and death of different types of cancer cells and found that the compound showed a pronounced suppression of colon and lung cancer cell growth, whereas suppression of pancreas, breast, ovarian and glioblastoma (GBM) cancer cells was more modest (Supplementary Fig. 6a). Interestingly, in four different normal cell lines, BC-LI-0186 and rapamycin showed little effect on cell growth and death, whereas 5-FU significantly affected both (Supplementary Fig. 6b, c).

**LRS as an effective target against rapamycin resistance**. Rapalogs, the first generation of mTOR inhibitors, often show limited anti-tumor efficacy due to acquired resistance; an understanding of the mechanisms involved, however, remains elusive[40, 41]. Many cancer-associated mutations that lead to mTORC1 hyperactivation have been found in genes encoding components of PI3K-AKT-mTORC1 pathway[42, 43] and *MTOR*[16]. To monitor the effect of BC-LI-0186 on cancer-associated *MTOR* mutations (Fig. 6a), we first determined the extent of their leucine dependency for S6K phosphorylation. We found that none of the mutants activated SK6 phosphorylation under 6 h leucine-starved conditions, whereas three of the mutants (L1406P, S2215Y, and I2500F) induced S6K phosphorylation under 1.5 h leucine-starved conditions (Fig. 6b), indicating they retained leucine dependency during mTORC1 activation. We then tested whether in cells expressing cancer-associated hyperactivating *MTOR* mutations, BC-LI-0186 could still inhibit the mTORC1 pathway through LRS interactions. We found that in SW620 cells, mTOR WT, L1460P (FAT domain mutant), and I2500F (FATC domain mutant) showed similar sensitivity to BC-LI-0186, rapamycin, and INK128 (mTOR kinase inhibitor) (Fig. 6c–e, Supplementary Fig. 7a, b). However, the rapamycin-resistant mutants, V2006L or F2108L[44], showed sensitivity to BC-LI-0186 and INK128, but not to rapamycin (Fig. 6f, g, Supplementary Fig. 7c, d). The kinase domain mutant S2215Y, which is known to be resistant to mTOR kinase inhibitor[18], showed sensitivity to BC-LI-0186 and rapamycin, but not to INK128 (Fig. 6h, Supplementary Fig. 7e). We also found that in cells expressing cancer-associated hyperactivating *MTOR* mutations, the growth inhibitory effects of BC-LI-0186, rapamycin, and INK128 were comparable to the inhibitory effects on S6K phosphorylation (Supplementary Table 2). These results indicate that BC-LI-0186 can inhibit the mTORC1 pathway via LRS even in the presence of the known cancer-associated *MTOR* mutations.

We further tested whether BC-LI-0186 could overcome acquired rapamycin resistance and inhibit the mTORC1 pathway, using two isogenic HCT116 cell lines that harbored either M*TOR* WT (HCT116 MW) or S2035I mutations (HCT116 MM)[20, 45]. We found that in HCT MW cells, rapamycin, BC-LI-0186, and INK128 all suppressed S6K phosphorylation in a dose-dependent manner (Fig. 7a, b). However, rapamycin-resistant HCT116 MM cells showed sensitivity to BC-LI-0186 and INK128, but not to rapamycin (Fig. 7a–c). We then treated all cell lines with BC-LI-0186 and rapamycin and compared their efficacies on cell proliferation and death. In HCT116 MW cells, rapamycin efficiently suppressed cell growth ($6.06 \pm 0.26$ nM $GI_{50}$) and induced cell death ($35.35 \pm 2.23$ nM $EC_{50}$). In HCT116 MM cells, however, its efficacy was significantly reduced ($167.53 \pm 1.28$ nM $GI_{50}$ and $330.49 \pm 16.27$ nM $EC_{50}$). In contrast, the efficacy of BC-LI-0186 changed little between the wild-type and mutant cells ($GI_{50}$: $39.49 \pm 2.74$ nM and $42.03 \pm 0.76$ nM, $EC_{50}$: $105.03 \pm 6.28$ nM and $100.45 \pm 2.73$ nM) (Fig. 7d and Supplementary Fig. 8a,

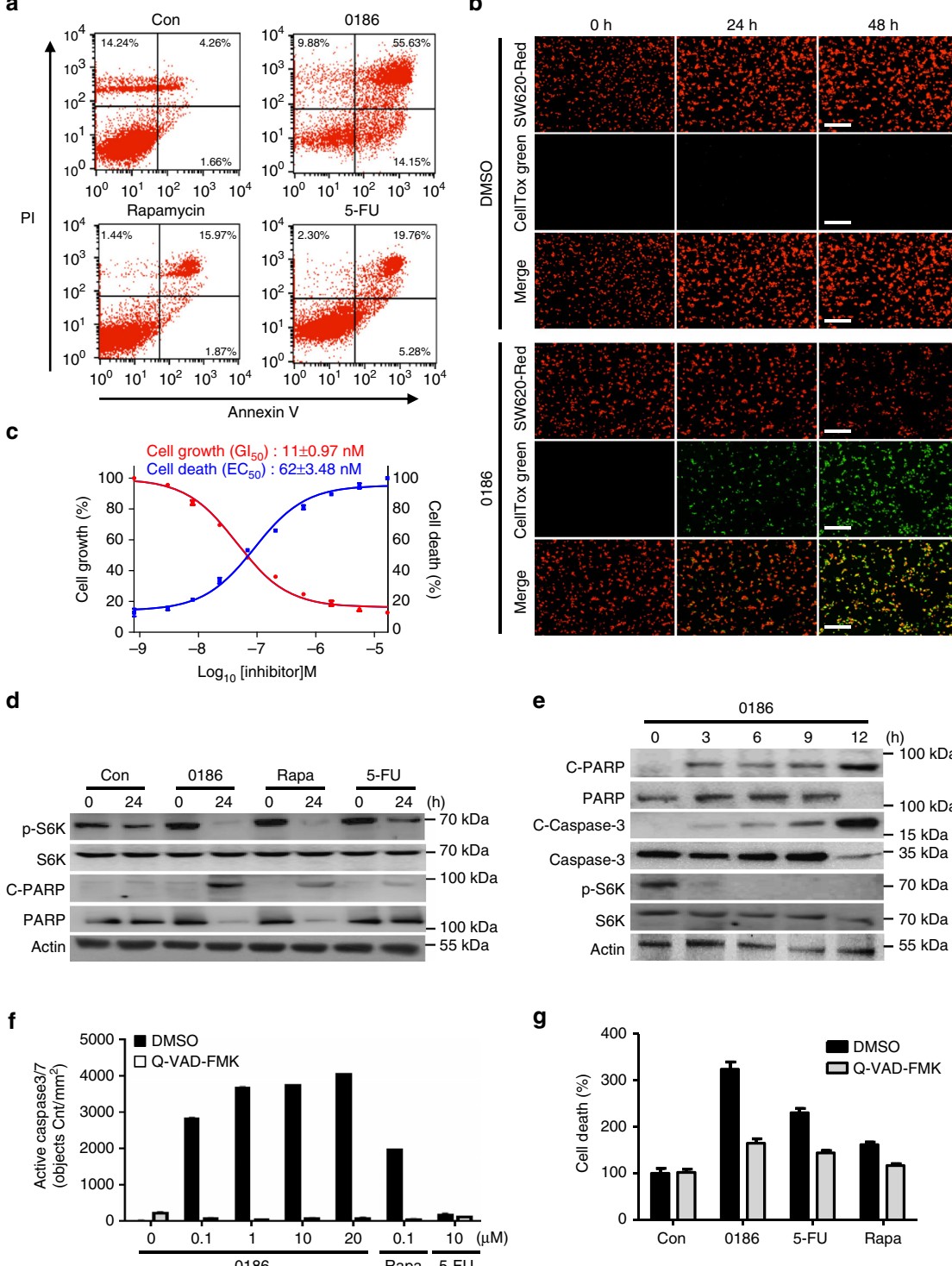

**Fig. 5** Effect of BC-LI-0186 on cancer cell death. **a** SW620 cells were treated with DMSO (Con), rapamycin (100 nM), BC-LI-0186 (10 μM), or 5-FU (10 μM) for 24 h. After cells were stained with PI and Annexin V, cells were separated and counted by FACS. **b** SW620 cells stably expressing RFP were treated with DMSO or 1.85 μM BC-LI-0186 in the presence of CellTox green. After 0, 24, 48 h, *red* (cell growth) and *green* (cell death) fluorescence images were acquired. **c** SW620 cells stably expressing RFP were treated with various concentration of BC-LI-0186 in the presence of CellTox green to monitor cell growth and death, simultaneously. The $GI_{50}$ (*red line*) and $EC_{50}$ (*blue line*) of BC-LI-0186 were determined by analyzing dose-response curves using GraphPad Prism tools. The *insets* represent the $GI_{50}$ and $EC_{50}$ values of BC-LI-0186. **d** SW620 cells were treated with DMSO (Con), BC-LI-0186 (10 μM), rapamycin (100 nM), or 5-FU (10 μM) for 24 h. Cells were analyzed by immunoblotting with the indicated antibodies. **e** SW620 cells were treated with 10 μM BC-LI-0186 for 0, 3, 6, 9, 12 h and analyzed by immunoblotting with the indicated antibodies. **f** SW620 cells were treated with DMSO (Con), BC-LI-0186 (0.1, 1, 10, 20 μM), rapamycin (100 nM), or 5-FU (10 μM). After 24 h, apoptosis was measured using Cellplayer caspase-3/7 reagent in the absence or presence of Q-VAD-FMK, which is an apoptosis inhibitor. **g** SW620 cells were treated with DMSO (Con), BC-LI-0186 (10 μM), rapamycin (100 nM), or 5-FU (10 μM) in the presence of CellTox green to measure cell death. After 24 h, green fluorescence was monitored in the presence of Q-VAD-FMK. The *error bars* represent mean ± S.D. ($n = 3$)

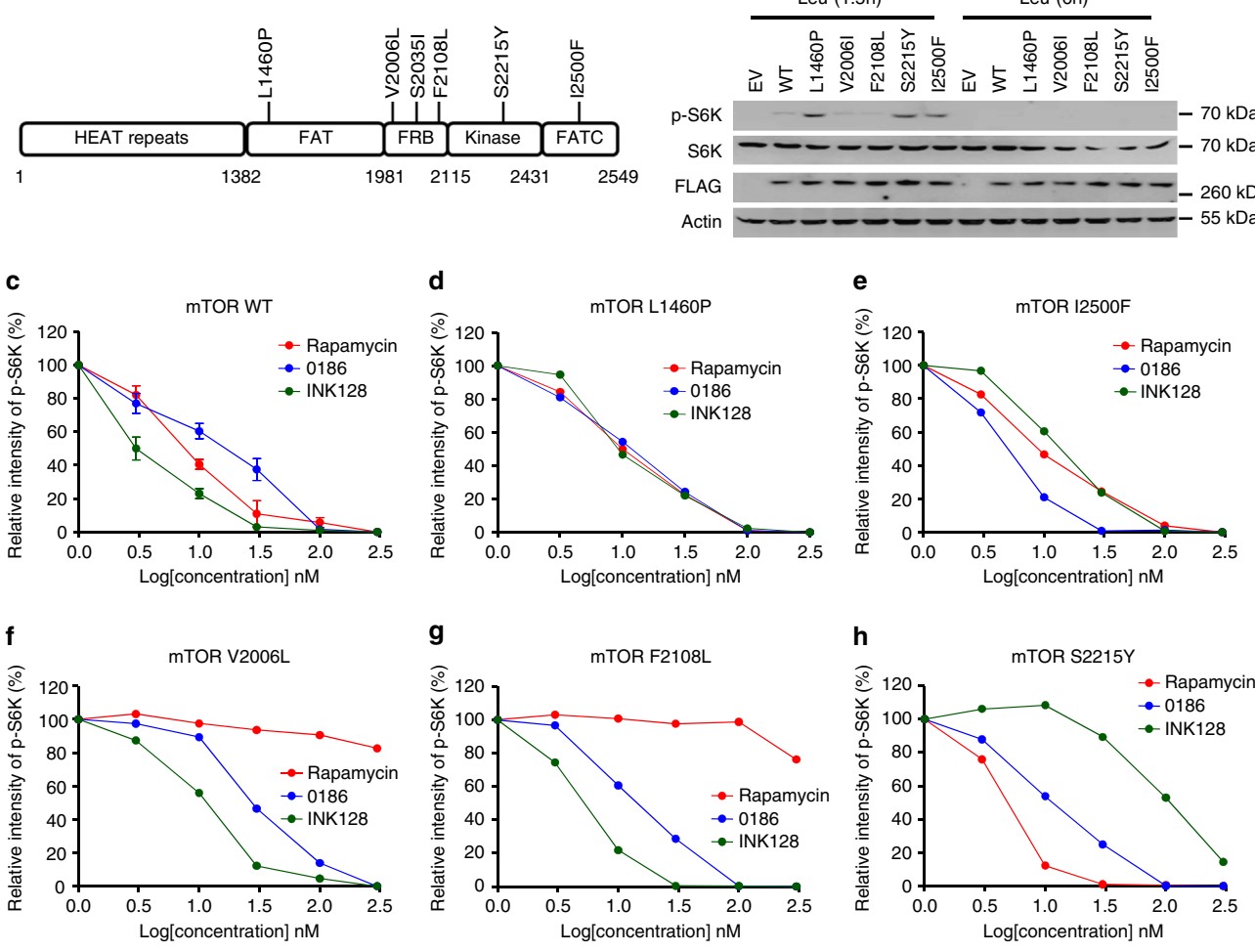

**Fig. 6** Effect of BC-LI-0186 on cancer-associated *MTOR* mutations. **a** Schematic representation of cancer-associated *MTOR* mutations. **b** Leucine-dependency of cancer-associated *MTOR* mutations. SW620 cells were transfected with FLAG-tagged mTOR WT and the indicated mutants. The cells were starved for leucine for 1.5 and 6 h, and the cell lysates were analyzed with the indicated antibodies. **c–h** Effect of BC-LI-0186 on S6K phosphorylation in SW620 cells harboring mTOR WT (**c**), mTOR L1460P (**d**), mTOR I2500F (**e**), mTOR V2006L (**f**), mTOR F2108L (**g**), mTOR S2215Y (**h**). The levels of p-S6K shown in Supplementary Fig. 7a–e were quantified and displayed as graph

b). In HCT116 MW and MM cells, BC-LI-0186 treatment suppressed GTP hydrolysis of RagD, whereas rapamycin and INK128 treatment showed no effect (Fig. 7e).

To validate the efficacy of BC-LI-0186 on HCT116 MM cells in vivo, we injected HCT116 MM cells into the subcutaneous right scapular region of nude mice. Intraperitoneal administration of BC-LI-0186 (20 mg/kg body weight) suppressed tumor growth by approximately 40% regardless of microsomal stability (Supplementary Table 3), whereas rapamycin was less effective (Fig. 7f and Supplementary Table 4). Together, these results suggest that targeting the leucine-sensing activity of LRS provides a strategy to overcome drug resistance caused by *MTOR* mutations in cancer.

## Discussion

Amino acid signaling is a mitogenic pathway that controls cell growth and metabolic processes. Although leucine is known to be the most effective amino acid for mTORC1 activation, glutamine and arginine can also activate mTORC1 via independent routes[11, 46, 47]. Our results provide validation that LRS has a crucial role as a leucine sensor for mTORC1 activation. Moreover, we show that there is a functional separation between LRS's ability to sense

leucine in mTORC1 activation pathway, and its ability to charge leucine onto cognate tRNAs.

In addition, this work is the first report on a specific chemical inhibitor of leucine-dependent mTORC1 signaling, BC-LI-0186. This compound binds to the VC domain of LRS, which is also a binding site for RagD (Figs. 2d, 3a), thereby preventing the mTORC1-activating interaction of LRS and RagD (Fig. 3e). The specific and unique activity of this compound was confirmed within the following contexts: its inhibition of leucine-dependent mTORC1 activation (Supplementary Fig. 1a); GTP hydrolysis of RagD (Fig. 4a, b); and lysosomal localization of mTORC1 (Supplementary Fig. 4c) without affecting the catalytic and editing activities of LRS (Supplementary Fig. 1c, e) and the kinase activity of mTOR (Supplementary Fig. 1b). These results suggest that LRS can coordinate amino acid metabolism through two different routes, namely, leucine charging to tRNA^Leu and leucine signaling to mTORC1 pathway. We further determined that leucine signaling inhibitor can control protein synthesis through specific inhibition of mTORC1.

Our and other studies have shown that LRS[14] and Sestrin2[26] mediate leucine signaling in mTORC1 pathway. Within the Rag GTPase cycle, however, they differ in their roles and mechanisms of action. Members of small GTP-binding proteins function as

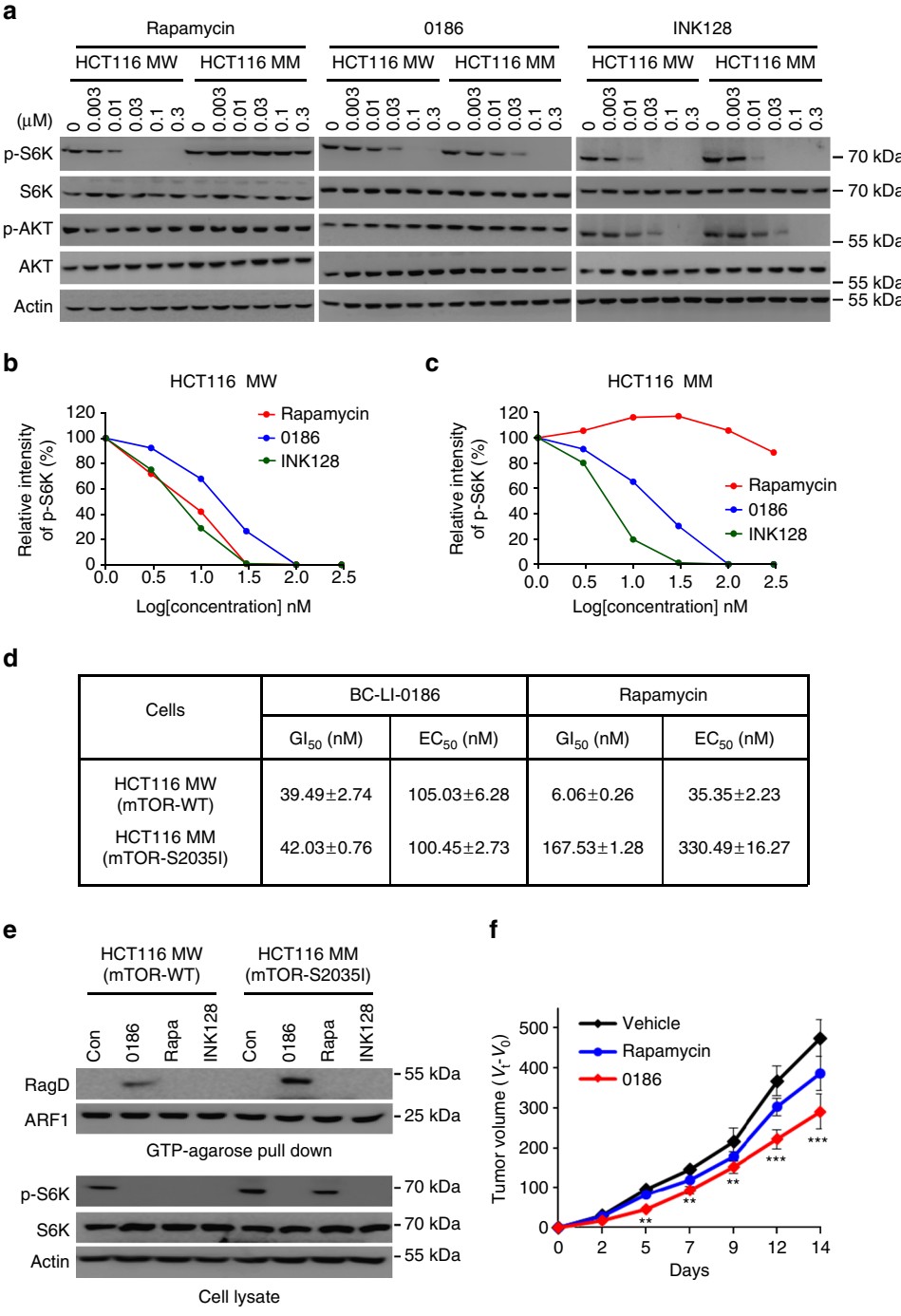

**Fig. 7** Efficacy of BC-LI-0186 to rapamycin-resistant cancer cells. **a** HCT116 MW (mTOR WT) and MM (mTOR S2035I) cells were treated with rapamycin, BC-LI-0186, and INK128 at the indicated concentrations (nM) for 6 h. The cell lysates were subjected to immunoblotting analysis with anti-mTOR, anti-p-S6K, anti-S6K, and anti-actin antibodies. **b** The level of p-S6K in HCT MW cells shown in **a** was quantified and displayed as graph. **c** The level of p-S6K in HCT MM cells shown in **a** was quantified and displayed as graph. **d** Cell growth $GI_{50}$ and cell death $EC_{50}$ values of BC-LI-0186 and rapamycin against HCT116 MW and MM cells. **e** GTP-agarose bead pull-down assays were used to monitor the GTP-bound RagD in HCT116 MW and MM cells. After cells were treated with 10 μM BC-LI-0186, 100 nM rapamycin, or 10 μM INK128, the cell lysates were incubated with GTP-agarose beads and the proteins precipitated with the beads were analyzed by immunoblotting with anti-RagD or anti-ARF1 antibodies. **f** HCT116 MM cells were injected subcutaneously to nude mice. BC-LI-0186 and rapamycin were intraperitoneally administered into the mice at the indicated doses every day (n/group = 6). Tumor volume was measured every other day for 2 weeks. The error bars represent mean ± S.D. *$p < 0.05$; **$p < 0.01$; ***$p < 0.001$ (vs. vehicle)

molecular GTP/GDP switches that cycle between active GTP-states and inactive GDP-states, and Rag GTPases form hetero-dimers that regulate mTORC1 activity[7]. GTP-bound RagA or RagB activates mTORC1, and GTP-bound RagC or RagD sup-presses it[7, 14]. Amino acid stimulation increases GTP-loading of

RagB[7]. Here, we show that leucine stimulation of cells increases GTP hydrolysis of RagD, whereas BC-LI-0186 treatment sup-presses this change (Fig. 4a, b). Together, these results suggest that at least two of the factors regulating GTP loading of RagA/B and GTP hydrolysis of RagC/D are coordinately required for

leucine-induced mTORC1 activation. Whereas LRS is a positive regulator of mTORC1 activation that serves a GAP role for RagD GTPase[14], Sestrin 2 is a negative regulator of mTORC1 that controls GTP hydrolysis of RagA/B through the regulation of GATOR1-GATOR2 pathway[48–51]. Hence, in the Rag GTPase cycle, LRS and Sestrin 2 function as "ON" and "OFF" switches, respectively. Here, we have shown that BC-LI-0186 specifically inhibits the LRS-RagD pathway by directly binding to LRS, but not Sestrin2 (Fig. 1e and Supplementary Fig. 1g), and specifically inhibits leucine-induced S6K phosphorylation (Supplementary Fig. 1a). How the several known regulators of Rag GTPases could work together to coordinate the Rag GTPase cycle needs further investigation.

Whereas LRS is normally bound to the multi-tRNA synthetase complex, its release from the complex allows it to interact with regulatory factors such as RagD[14] and Vps34[32]. Interestingly, LRS appears to use distinct regions for the interactions with different partners. For instance, BC-LI-0186 blocked the LRS–RagD interaction while giving no effects on the interactions between LRS and EPRS or between LRS and Vps34 (Fig. 3d).

Amino acids promote the loading of RagB with GTP, which provides the docking site for mTORC1 on lysosomal membrane through the direct interaction with Raptor[7, 10]. In addition, LRS is known to control lysosomal translocation of mTORC1 in a leucine-dependent manner[14]. Here, we show that a leucine signaling inhibitor blocks lysosomal translocation of mTORC1. These results suggest that it may be possible to inhibit mTORC1 by controlling its lysosomal localization, and this may be accomplished independent of targeting MTOR mutations. Indeed, isogenic HCT116-human colon cancer cells expressing wild-type (HCT116 MW) and rapamycin-resistant S2035I mutants of MTOR (HCT116 MM) showed a similar sensitivity to the leucine signaling inhibitor, whereas HCT116 MM cells showed a reduced sensitivity to rapamycin (Fig. 7a, d). Thus, the leucine-sensor function of LRS represents a novel therapeutic target by which to suppress cancers, especially those that have acquired resistance to rapamycin. It would be interesting to see whether this function would be also applicable to control other diseases associated with hyperactive mTORC1 signaling.

## Methods

**Materials**. Antibodies used in this study can be found in Supplementary Table 5. The secondary antibodies such as goat anti-mouse AlexaFluor 568, goat anti-rabbit AlexaFluor 568, goat anti-mouse Alexa Fluor488, and goat anti-rabbit AlexaFluor 488 from Molecular probes. GTP-agarose, actin antibodies were from Sigma-Aldrich. An enhanced chemiluminescence kit (ELC) system was from Thermo Fisher Scientific. CellTox green and DOX were from Promega. Leucine-free DMEM, glutamine-free DMEM, and arginine-free DMEM were from Welgene and Invitrogen. Rapamycin and INK128 were from Calbiochem.

**Cell culture**. SW620, COLO201, WiDr, HCT-15, SW480, DLD-1, HCT-116, HT-29, LS1034, LoVo, NCI-H508, COLO205, LS174T (Human colon cancer cells), NCI-H1975, A549, NCI-H226, NCI-H1793, NCI-H1650, NCI-H596, NCI-H460, NCI-H1299, NCI-H358, HCC-44, HCC-2108 (Human lung cancer cells), Panc-1, MIA-Paca-2, BxPC-3, AsPC-1, Panc10.05 (Human pancreatic cancer cells), MDM-MB-231, HCC1569, HCC70, HCC1937, HCC1395 (Human breast cancer cells), NIH:OVCAR-3, SK-OV-3, Caov-3, SNU119, UWB1.289 (Human ovary cancer cells), U343, T98G, SNB-19, A-172, U87 (Human glioblastoma cells), FHC (human normal colon epithelial cell), WI-26 (human normal lung fibroblast), NIH3T3 (mouse normal fibroblast), BJ (human normal foreskin fibroblast) cells were purchased from American Type Culture Collection (ATCC) or Korean Cell Line Bank. All cells were cultured in appropriated medium supplemented with 10% heat inactivated fetal bovine serum and 1% penicillin-streptomycin at 37 °C in a 5% CO2. HCT116 (mTOR-WT) and isogenic HCT116 (mTOR-S2035I-Homo) cells were purchased from Horizon were cultured in RPMI supplemented with 10% fetal bovine serum and 1% penicillin-streptomycin. DOX-inducible SW620 stable cell lines expressing either LRS shRNA or myc-LRS were generated. These cell lines were selected with antibiotics, puromycin (1 μg/ml) for 4 weeks. All cell lines were tested for mycoplasma contamination.

**Tet-On inducible lentivirus production and infection**. The vector utilizes the Tet-On 3G induction system. In the presence of DOX, the shRNA or mRNAs were constitutively expressed by the TRE3G promoter. The SMART choice inducible lentivirus either encoding shRNA directed against LRS (VSH6376-220936786, 5′-CTGGACATCACTTGTTTCT-3′) or scrambled control shRNA were purchased from Dharmacon (Thermo Scientific). These lentiviruses were infected into SW620 cells of 60-mm dish in the presence of 8 μg/ml polybrene (Sigma-Aldrich) within the serum-free RPMI. After 12 h infection, the complete medium were replaced in the presence of puromycin (1 μg/ml). To generate the Tet-On inducible LRS lentivirus, myc-LRS was subcloned to pLVX-TetOne vector. All procedures were slightly modified according to the manufacturer's instructions (Clontech). Both myc-LRS plasmid and lenti-X HTX packing mix 2 were transfected to HEK-293T cells with Xfect Polymer. After 5 h, the transfection medium were replaced with fresh complete growth medium. Lentivirus collected 2 days after transfection was filtered through a 0.45 μm filter to remove cellular debris. Lentivirus was infected into SW620 cells in the presence of 8 μg/ml polybrene (Sigma-Aldrich). After 5 h, SW620 cells were selected with RPMI supplemented with 10% FBS and 1% PS in the presence of puromycin (1 μg/ml) for 4 weeks.

**In vivo GTPase assay**. In vivo GTPase assay was done as previously described[14]. Briefly, SW620 cells were washed with phosphate-free DMEM two times and further incubated with phosphate-free DMEM for 60 min. After labeling with 100 μCi of [$^{32}$P]phosphate/ml for 8 h, cells were lysed with lysis buffer (50 mM Tris, pH 7.5, 100 mM NaCl, 10 mM MgCl2, 1 mM DTT, 0.5% NP-40, protease inhibitor cocktail) for 30 min on ice. Cell lysates were centrifuged at 12,000 × g for 15 min at 4 °C. To quench GAP activity, the supernatant (160 μl) was mixed with 500 mM NaCl (16 μl). HA-RagD or HA-ARF1 was immunoprecipitated with anti-HA antibody for 1 h at 4 °C. The beads were washed with buffer 1 (50 mM Tris, pH 8.0, 500 mM NaCl, 5 mM MgCl2, 1 mM DTT, 0.5% Triton X-100) three times at 4 °C and then washed with buffer 2 (50 mM Tris, pH 8.0, 100 mM NaCl, 5 mM MgCl2, 1 mM DTT, 0.1% Triton X-100) three times at 4 °C. The HA-RagD-bound or HA-ARF1-bound nucleotides were eluted with 20 μl of buffer 3 (2 mM EDTA, 0.2% sodium dodecyl sulfate, 1 mM GDP, 1 mM GTP) at 85 °C for 3 min. The eluted nucleotides were separated onto polyethyleneimine cellulose plates (Baker-flex) in 0.75 M KH2PO4[pH 3.4] solution. GTP and GDP was visualized and quantified by a phosphoimager.

**GTP-agarose bead pull-down assay**. The GTP loading activity of RagD GTPase was monitored by pull-down assay using GTP-agarose beads[35–38]. Specific pull down of GTP-loaded RagD was validated first using myc-tagged RagD GTP mutant (Q121L) or GDP mutant (S77L). After cells were rinsed in ice-cold PBS, they were collected in GTP-binding buffer (20 mM Tris-HCl, pH 7.5, 5 mM MgCl2, 2 mM PMSF, 20 μg/ml leupeptin, 10 μg/ml aprotinin, 150 mM NaCl, 1% Triton X-100, 1× phosphatase inhibitor cocktail). Cells were lysed by sonication for 15 s. The lysates were then centrifuged at 13,000 × g for 10 min at 4 °C, and the supernatant was collected. Protein extracts were incubated with 100 μl GTP-agarose beads (Sigma-Aldrich, cat no. G9768) in a total of 500 μl of GTP-binding buffer for 30 min at 4 °C. The beads were washed with GTP-binding buffer, and the supernatant was retained. Then retained supernatant was incubated with washed beads for another 30 min. The beads were washed again, and then incubated with the retained supernatant overnight at 4 °C. After washing five times with GTP-binding buffer to further dissociate other potentially bound contaminants, the GTP-bound proteins were eluted and visualized by performing immunoblot analysis.

**SAXS data collection and analysis**. The human full-length LRS gene was cloned in the vector pET16b (Novagen) with an N-terminal His9-tag. The recombinant human LRS was expressed in bacterial strain E. coli Rosetta (DE3) and purified by a tandem purification approach, using a 5 ml Ni-HiTrap affinity column, a home-packed Sepharose Q column and a gel filtration column Superdex 200 (10/300) (GE Healthcare) in turn. The final protein was eluted from the gel filtration column with Buffer A (150 mM NaCl, 25 mM Hepes-Na at pH 7.5 and 5% glycerol) and the peak fraction of the sample was used without further concentration to avoid aggregation. For each sample group three serial dilutions were prepared with highest and lowest concentrations of 4.5 and 1.1 mg/ml, respectively. After elution from gel filtration column, the samples were flash frozen with liquid nitrogen and stored frozen. Before frozen Leu-AMS or BC-LI-0198 were added to the corresponding samples with final concentrations 2-fold of the highest concentrations of LRS protein. The SAXS data were collected at SIBYLS beamline 12.3.1 with two short exposures (0.5 s and 1 s) and one long exposure (6 s) for each sample at room temperature[52]. SAXS data were evaluated, scaled, and merged by PRIMUS. Evaluation of pair distance distribution function and determination of the value of the maximum diameter ($D_{max}$) of the particle were carried out with GNOM by checking the fit to the experimental data curves. DAMMIF slow mode was used for ab initio modeling and for each sample 40 independent simulations were performed. The final averaged shape models were obtained from superposition, averaging and filtering by DAMAVER program. The rigid three-dimensional structure models were then docked in these SAXS envelopes manually with Pymol and Chimera.

**Three-dimensional structure modeling.** The prototype model of human LRS was generated by submitting the amino acid sequence of human full-length LRS gene to Phyre2 online server[53]. The structure model of CP1 domain, catalytic domain (CD), anticodon-binding domain (ABD) and VC domain was obtained using crystal structure of *Pyrococcus horikoshii* LRS with tRNA complex (PDB 1WZ2B) as model template with 32% sequence identity and 89% sequence coverage. The CP1 domain model was then replaced by the crystal structure of human LRS CP1 domain (PDB 2WFDA) by superimposition using the pairwise DaliLite server. The model prediction of the C-terminal UNE-L domain was performed by Robetta online Rosetta server using chain A of linear ubiquitin and antibody Fab fragment complex crystal structure as template (PDB 3U30A). The structural model prediction for the CD, ABD, and VC domains were based on templates c1qu2A, d1ivsa2, and c1wzB, respectively (sequence identity 21%). In total, 463 residues (94% sequence coverage), 126 residues (89% sequence coverage), and 135 residues (79% sequence coverage) were modeled for the three domains with confidence values of 100, 100, and 98.5%, respectively.

**HDX FT-ICR MS data collection and analysis.** The LRS protein samples for HDX-MS assays were expressed and purified using the same methods as those for SAXS except being eluted from the gel filtration column with Buffer B (Buffer A without 5% glycerol). The concentrations of LRS protein of LRS apo, with BC-LI-0198, and with Leu-AMS/BC-LI-0198 were 19, 33, and 33 μM, respectively. The final concentrations of Leu-AMS and BC-LI-0198 in the samples were 2-fold of those of LRS protein. After incubation, these samples were flash frozen in liquid nitrogen and stored frozen. The HDX methods were described previously[54]. Briefly, when HDX reactions were initiated, 5 μl of a sample was diluted to 45 μl of 150 mM NaCl, 25 mM Hepes-Na at pH 7.5 in $D_2O$. For the blank control, the dilution was done with buffer in $H_2O$. Reactions were performed in triplicate at 1–2 °C to minimize back exchange with incubation periods of 0.5, 1, 2, 4, 8, 15, 30, 60, 120, and 240 min, each followed by quench and proteolysis by adding protease type XIII (Sigma Aldrich) solution in 1.0% formic acid. The samples were incubated in protease solution 3 min at 1 °C. Peptide separation and desalting were performed over a Pro-Zap Expedite MS C18 column (1.5 μm particle size, 500 Å pore size, $2.1 \times 10$ mm[2]; Grace Davidson) with a Jasco HPLC/SFC (high-performance liquid chromatography/supercritical fluid chromatography) system. The peptides were eluted by a fast gradient from 2 to 95% B (A: acetonitrile/$H_2O$/formic acid, 5/94.5/0.5; B: acetonitrile/$H_2O$/formic acid, 95/4.5/0.5) in 3 min to minimize back exchange. Peptides were ionized by electrospray ionization (ESI) and introduced to 14.5 tesla FT-ICR MS (Fourier Transform Ion Cyclotron Resonance Mass Spectrometry) with an LTQ Velos front end, with high mass resolving power ($m/\Delta m_{50\%} = 200,000$ at $m/z$ 400). Data were analyzed by a custom-built Predator software package[55].

**Molecular docking analysis.** The method employs a cavity detection algorithm and a shape comparison filter is combined with a Monte Carlo conformational search for generating ligand poses consistent with the active site shape. The top 10 conformations were generated based on the DockScore value after the energy minimization using smart minimizer method. The docked poses were then analyzed using the scoring functions, including Dock scores, LigScore1 and LigScore2, PLP1 and PLP2, Jain, and potential of mean force scores. Highly scored 50 ligand conformations were archived based on the scoring functions. Next, the in situ minimization of ligand conformation was performed on the selected configuration to minimize both the ligand internal energy and the ligand–receptor interaction energy while holding the receptor atom coordinates fixed.

**Immunoblot analysis.** Proteins were denatured by boiling at 95 °C for 5 min in a Laemmli sample buffer. The denatured proteins were then separated by sodium dodecyl sulfate-polyacrylamide gel electrophoresis and transferred to nitrocellulose membranes. After blocking in TTBS buffer (10 mM Tris-HCl, pH 7.5, 150 mM NaCl, 0.05% Tween 20) containing 5% skimmed milk powder, the membranes were incubated with individual monoclonal or polyclonal antibodies and subsequently re-incubated with either anti-mouse or anti-rabbit IgG that were coupled with horseradish peroxidase. Detection was performed using an enhanced chemiluminescence kit, according to the manufacturer's instructions. Uncropped images of all western blots presented in the main figure are provided in Supplementary Fig. 9.

**In vitro pull-down assay.** GST or GST-fused RagD proteins were expressed in BL21 *E. coli*, lysed with lysis buffer, and purified by glutathione sepharose 4B. GST or GST-fused RagD proteins were incubated with purified His-LRS in the absence or presence of BC-LI-0186 for 2 h and then pulled down with glutathione sepharose 4B. Binding assay was conducted in 50 mM Hepes/NaOH (pH 7.4), 150 mM NaCl, 5 mM $MgCl_2$, 0.5 mM EDTA, and 0.1% Triton X-100.

**Surface plasmon resonance.** The dissociation rate constant (KD) toward His-LRS WT or His-Sestrin2 with BC-LI-0186 and BC-LI-0198 or toward His-LRS S974A with BC-LI-0186 were determined using Biacore T-200 (GE Healthcare). Each protein was diluted and adjusted to 30 μg/ml in 10 mM sodium acetate buffer (pH 5.0). The carboxyl group on the surface of CM5 sensor chip was replaced with

reactive succinimide ester using combination of 1-ethyl-3-(3-dimethylaminopropyl)-carbodiimide and N-hydroxysuccinimide (NHS). BC-LI-0186 and BC-LI-0198 were diluted in PBS with 0.05% (v/v) Tween 20 and 2% (v/v) DMSO and injected at a flow rate of 20 μl/min at 25 °C, and the binding was determined by the change in resonance units. Final sensorgram was obtained after the elimination of response from buffer-only control. The dissociation constant (KD) was calculated by fitting the sensorgram to the 1:1 binding model. Data analysis was done by using Biacore T-200 Evaluation software.

**Subcellular fractionation.** Cells were washed, harvested, and lysed. Lysosome fraction was obtained using lysosome enrichment kit (Thermo Fisher Scientific) following the manufacturer's instruction with minor modification. Briefly, cells were lysed with Dounce homogenizers on ice. After centrifugation of cell lysates at $500 \times g$ for 10 min, the supernatant was prepared with discontinuous density gradient including 30, 27, 23, 20, and 17% gradient. After the ultracentrifugation of supernatant at $145,000 \times g$ for 2 h at 4 °C, the lysosome band was collected in the top band of the gradient.

**Tumor xenograft in athymic nude mice.** All animal experiments were conducted in accordance with the Guide for the Care and Use of Laboratory Animals, published by Korea National Institute of Health (BEC-CAN-2016-001 and 002). Xenografts were performed on 8-week-old Balb/c female nude mice. After the subcutaneous injection with $1.2 \times 10^7$ cells in 300 μl serum-free RPMI medium, tumor volumes were measured by electronic calipers at every other day and calculated using the formula (length × width × height)/2. Mice were randomized by tumor size and treatment was initiated when tumors reached an average size of 50–100 mm³. BC-LI-0186 was dissolved in vehicle solution (10% DMAC, 10% Tween 80, 80% saline), and was administrated (TID at 4 h intervals, 20 mg/kg, intraperitoneal injection). All evaluators were blinded to treatment groups.

**Leucylation assay.** The leucylation assay was carried out in a buffer containing 50 mM HEPES-KOH (pH 7.6), 25 mM KCl, 5 mM $MgCl_2$, 1 mM spermine, 5 mM ATP, 2 mg/ml tRNA (yeast extract), [³H] Leucine (60 Ci/mmol), and 50 nM cytosolic LRS or 2.5 μM mitochondrial LRS (LRS2). Reactions were initiated with enzyme and conducted in a 37 °C heat block. At 10 min (cytosolic LRS) or 30 min (LRS2), aliquots (15 μl) were taken and quenched on Whatman filter pads (cat. no. 1003323, grade 3, 2.3 cm) that were presoaked with 5% trichloroacetic acid (TCA). The pads were washed three times for 10 min each with cold 5% TCA and once with cold 100% ethanol. The washed pads were then dried. Put it into LS vial (WHEATON-986701) with 5 ml of the cocktail solution. Radioactivity was quantified in a scintillation counter (Perkin Elmer).

**ATP-PPi exchange assay.** The ATP-PPi exchange reaction was performed in a reaction mixture containing 2 mM [³²P]pyrophosphate (PPi) (80.70 mCi/ml), 50 mM Hepes-KOH (pH 7.6), 2 mM $MgCl_2$, 8 mM KF, 4 mM ATP, 10 mM leucine, and 25 nM LRS. Reaction was initiated with enzyme and conducted in 37 °C heat block for 5 min. Aliquots (10 ml) were taken at 10 min point and the reaction was stopped using 1 ml quenching buffer (50 mM NaPPi, 3.5% $HClO_4$, 2% activated charcoal). The charcoal suspension was filtered through a Whatman GF/A filter, washed three times with 10 ml of water, and rinsed with 10 ml of 100% ethanol. The charcoal powder on the filters was dried, and the synthesized [³²P]ATP was counted using a scintillation counter (Beckman Coulter).

**Editing activity assay.** Transcribed tRNA^Leu at a final concentration of 20 μM was misaminoacylated with 1 μM editing-defective *Thermus thermophilus* LRS (D347A) in a reaction mixture containing 50 mM HEPES-KOH (pH 7.6), 25 mM KCl, 5 mM $MgCl_2$, 1 mM spermine, 5 mM ATP, and 40 μM [³H]Met, yielding mischarged tRNA^Leu. The reactions were conducted at a 37 °C heat block for 60 min. The mixtures were isolated by acid phenol/chloroform extraction (pH 4.5) following by isopropanol precipitation to obtain mischarged tRNA^Leu. Deacylation assay was carried out at 37 °C in a 35 μl reaction mixture containing 50 mM HEPES-KOH (pH 7.6), 25 mM KCl, 5 mM $MgCl_2$, 1 mM spermine, 5 mM ATP, and 2.5 μM [³H]Met-t tRNA^Leu, 50 nM human LRS, and indicated concentrations of BC-LI-0186 or AN2690. Aliquots (15 μl) were taken at the indicated time points and quenched on Whatman filter pads that were presoaked with 5% TCA. The pads were washed and analyzed as described above.

**Immunofluorescence staining.** Cells were seeded onto coverslips and fixed with 4% paraformaldehyde for 2 h. After the incubation of PBS blocking solution containing 2% goat serum, the cells were incubated with primary antibody for 4 h and Alexa 488 or Alexa 594-conjugated secondary antibody for 1 h in PBS blocking solution. Nuclei were stained with DAPI and lysosome was stained with LAMP2. After washing with PBS containing 0.1% Triton-X 100, the cells onto coverslips were mounted and observed with confocal laser scanning microscope (Nikon, A1R). To reduce the background via removing of cytosolic proteins, cells were treated with 25 μg/ml digitonin for 5 min on ice.

**Thermal shift assays**. LRS samples used in Thermal shift assays were purified with the same methods as those used in HDX-MS assays. After adding BC-LI-0198, the volumes of the mixtures were adjusted to 10 μl by water. Then the samples were incubated on ice for 30 min followed by mixing the samples with the same volume of 2 × Protein Thermal Shift Dye (Applied Biosystems, Life Technologies) solution. The final concentrations of LRS and BC-LI-0198 were 2 and 4 μM, respectively. The assays were performed on an iQ5 RT-PCR instrument (Bio-Rad) with the method of setting 20 and 90 °C as start and end temperatures, respectively, and increasing 1 °C per minute. For each sample, six copies were carried out at the same time and the final curves were averaged from those of the six copies.

**Measurement of mTORC1 activity**. For measurement of leucine-dependent mTORC1 activity, cells were rinsed with leucine-free DMEM twice, and incubated in leucine-free DMEM for 90 min, and stimulated with leucine-containing DMEM for 15 min. For measurement of glutamine-dependent or arginine-dependent mTORC1 activity, cells were starved glutamine or arginine for 90 min and then stimulated with glutamine-containing or arginine-containing DMEM for 30 min. For measurement of mTORC1 activity, cells were treated with compounds for 6 h. After cells were rinsed in cold PBS, cells were lysed on 4 °C shaker for 90 min in lysis buffer (50 mM Tris-HCl pH 7.4, 5 mM EDTA, 150 mM NaCl, 10% glycerol, 0.5% Triton X-100, protease inhibitor cocktail, and phosphatase inhibitor cocktail). Cell lysates were analyzed by immunoblotting with phospho-S6K (T389) antibody (Cell Signaling, #9205) or phospho-4EBP1 (T37/46) antibody (Cell Signaling, #9495). Relative intensity of p-S6K/actin was used to monitor mTORC1 activity. $IC_{50}$ values were calculated with Graph Pad Prism.

**Cell growth and viability assays**. SW620 cells stably expressing red fluorescence were designed by using CellPlayer[TM] NuCLight Red (Essen BioScience, Cat no. 4476). Red SW620 cells were seeded in 96-well plates, incubated for 24 h, and treated with compounds diluted in the media containing CellTox[TM] Green Dye (Promega). Phase and fluorescence images were acquired every 12 h using Incu-Cyte[TM] Zoom (Essen BioScience). Quantitative analysis was performed by using IncuCyte[TM] Zoom basic analyzer. Green color (CellTox[TM] object count/mm$^2$ over time) was used to quantify number of dead cells. Red object count/mm$^2$ over time is used to quantify number of cell viability. The growth or inhibition curves, $IC_{50}$ and $EC_{50}$ values were calculated with Graph Pad Prism.

**Normal cell viability**. Human colon normal epithelial cells (ATCC) were seed in 96-well plates. After 24 h, cells were incubated with compounds for additional 24 h. Compounds were diluted in the media containing CellTox[TM] Green Dye (Promega). Quantification of cell viability was performed by using IncuCyte[TM] Zoom basic analyzer. Viable cells or Green objects were counted to quantify cell growth or death.

**Solubility assay**. The thermodynamic solubility of analogs was determined by using a HPLC and quantified by a calibration curve generated from the HPLC analysis of each analog. For the generation of calibration curve, each compound was dissolved completely in methanol or DMSO at the concentration of 500 μg/ml. The analytic samples were prepared from the 3-fold serial dilution by six times and those were used for the HPLC assay after filtration through a hydrophilic GHP membrane with 0.2 μm pore size. Then, 1.5 mg of each analog samples were added to approximately 1.5 ml of distilled water or PBS to make saturated solutions, sonicated for 15 min, and shaken horizontally for 30 min. The samples were then filtered through a GHP membrane and analyzed using a HPLC. The assay was performed on a Shimadzu LC-20A HPLC system using an Agilent XDB-C$_{18}$ column (150 × 4.6 mm i.d.; particle size 5 μm, Agilent Technologies). The detection wavelength was 254 nm. The mobile phase was 10–20%/0.05% formic acid in H$_2$O/0.05% formic acid in acetonitrile at the flow rate of 1 ml/min. The injection volume was 3 μl.

**In silico physicochemical profiles**. The AlogP and Molecular_PolarSurfaceArea were calculated by Discovery Studio 4.5 (Accelrys Software, Inc., San Diego).

**Preparation of BC-LI-0186**. All chemicals were obtained from commercial suppliers and used without further purification. All reactions were monitored by thin-layer chromatography (TLC) on pre-coated silica gel 60 F$_{254}$ (mesh) (E. Merck, Mumbai, India), and spots were visualized under UV light (254 nm). Flash column chromatography was performed with silica (Merck EM9385, 230–400 mesh). $^1$H and $^{13}$C nuclear magnetic resonance (NMR, Agilent Technologies) spectra were recorded at 400 and 100 MHz. Proton and carbon chemical shifts are expressed in ppm relative to internal tetramethylsilane, and coupling constants ($J$) are expressed in Hertz. Splitting patterns were presented as s, singlet; d, doublet; t, triplet; q, quartet; dd, double of doublets; m, multiplet; br, broad. LC–MS (liquid chromatography-mass spectrometry) spectra were recorded by ESI probe using a Shimadzu LCMS-2010 instrument with an Agilent C$_{18}$ column (50*4.6 mm, 5 μm). The detected ion peaks were ($M^+z$)/$z$ in positive, where $M$ represents the molecular

weight of the compound and $z$ represents the charge (number of protons). High-resolution ESI-MS measurements were performed on an Agilent 6530 Accurate-Mass Quadrupole Time-of-Flight (Q-TOF) LC/MS system (Santa Clara, CA, USA) at Yonsei University; positive mode. To determine the purity of the final compounds, HPLC experiments were conducted using agilent analytic column eclipse-XDB-C$_{18}$ (150*4.6 mm, 5 μm) on Shimadzu HPLC-2010 (HPLC) instruments. BC-LI-0186 (4-(4-isopropyl-2,3-dimethyl-5-oxo-2,5-dihydro-1H-pyrazol-1-yl)-N-(phenoxymethyl)benzenesulfonamide) was purchased from Vitas-M Laboratory, Catalog #STK736938, > 99% HPLC purity.

TLC (ethyl acetate:hexane, 1:1 v/v): $R_f = 0.26$; ESI (m/z) 430 (MH$^+$) 452 (MNa$^+$) 428 (MH$^-$); $^1$H NMR (400 MHz, DMSO-d6): δ 7.97 (s, 1H), 7.91 (d, $J = 8.4$ Hz, 2H), 7.56 (d, $J = 8.8$ Hz, 2H), 7.25 (t, $J = 8.0$ Hz, 2H), 6.91 (t, $J = 7.4$ Hz, 2H), 6.83 (d, $J = 8.0$ Hz, 2H), 3.95 (t, $J = 5.4$ Hz, 2H), 3.18 (d, $J = 4.8$ Hz, 2H), 2.95 (s, 3H), 2.81–2.71 (m, 1H), 2.21 (s, 3H), 1.20 (d, 6H, $J = 6.8$ Hz); $^{13}$C NMR (100 MHz, DMSO-d6): δ 165.2, 158.5, 155.5, 139.1, 136.9, 129.8, 127.9. 122.1, 121.1, 114.7, 114.5, 66.6, 42.4, 37.1, 23.9, 21.3, 11.2; HRMS (ESI) [M+H$^+$] calculated for C$_{22}$H$_{27}$N$_3$O$_4$S: 430.1798, found: 430.1795 (Supplementary Fig. 10).

**Synthesis of BC-LI-0198**. Scheme of chemical synthesis of BC-LI-0198 (N-(3,4-dimethoxybenzyl)-4-(4-isopropyl-2,3-dimethyl-5-oxo-2,5-dihydro-1H-pyrazol-1-yl)benzenesulfonamide) is available in Supplementary Information (Supplementary Fig. 11).

**4-(4-Isopropyl-2,3-dimethyl-5-oxo-2,5-dihydro-1H-pyrazol-1-yl)benzene-1-sulfonyl chloride (1)**. To a round bottom flask containing neat chlorosulfuric acid (43.3 ml, 651 mmol) at 0 °C was added 4-isopropyl-1,5-dimethyl-2-phenyl-1H-pyrazol-3(2H)-one (15 g, 65.1 mmol) portionwise. The solution was then heated to 40 °C and stirred for 72 h. After cooling to room temperature, the reaction mixture was then poured slowly into ice water. Dichloromethane (100 ml) was added and the organic layer was separated and dried over MgSO$_4$. The solution was concentrated in vacuo and purified by column chromatography (100% dichloromethane) to give 4-(4-isopropyl-2,3-dimethyl-5-oxo-2,5-dihydro-1H-pyrazol-1-yl)benzene-1-sulfonyl chloride (11.3 g, 53% yield) as a light yellow solid.

$^1$H NMR (400 MHz, DMSO-d6): δ 8.07 (d, $J = 8.8$ Hz, 2H), 7.71 (d, $J = 8.8$ Hz, 2H), 2.97 (s, 3H), 2.83–2.72 (m, 1H), 2.20 (s, 3H), 1.26 (d, $J = 7.2$ Hz, 6H); $^{13}$C NMR (100 MHz, DMSO-d6): δ 165.8, 155.5, 141.7, 139.6, 128.2, 121.6, 117.2, 37.6, 24.3, 20.8, 11.4.

**N-(3,4-dimethoxybenzyl)-4-(4-isopropyl-2,3-dimethyl-5-oxo-2,5-dihydro-1H-pyrazol-1-yl)benzenesulfonamide (BC-LI-0198) (2)**. To a round bottom flask containing veratrylamine (91.53 mg, 0.547 mmol) was added neat sulfonyl chloride compound (1, 150 mg, 0.456 mmol) and pyridine (216.50 mg, 2.737 mmol) with dichloromethane (2 ml). The solution was then stirred for 3 h at room temperature. The solution was concentrated in vacuo and purified by column chromatography (ethyl acetate:hexane = 1:1) followed by recrystallization using dichloromethane and hexane to give the compound BC-LI-0198 (209.64 mg, 50.66% yield) as a light pink solid.

TLC (ethyl acetate:hexane, 1:1 v/v): $R_f = 0.31$; ESI (m/z) 460 (MH$^+$) 482 (MNa$^+$) 458 (MH$^-$); $^1$H NMR (400 MHz, DMSO-d6): δ 8.10 (t, $J = 6.8$ Hz, 1H), 7.83 (d, $J = 8.4$ Hz, 2H), 7.51 (d, $J = 8.4$ Hz, 2H), 6.81–6.72 (m, 3H), 3.96 (d, $J = 6.0$ Hz, 2H), 3.68 (s, 3H), 3.65 (s, 3H), 2.96 (s, 3H), 2.81–2.70 (m, 1H), 2.22 (s, 3H), 1.19 (d, $J = 6.8$ Hz, 6H); $^{13}$C NMR (100 MHz, DMSO-d6): δ 165.1, 155.5, 148.8, 148.3, 138.9, 137.2, 130.0, 127.9, 122.0, 120.2, 114.5, 111.8, 111.7, 55.8, 55.7, 49.0, 46.5, 37.1, 23.9, 21.3, 11.2; HRMS (ESI) [M+H$^+$] calculated for C$_{23}$H$_{29}$N$_3$O$_5$S: 460.1911, found: 460.1901 (Supplementary Fig. 12).

**Statistical analysis**. The comparisons of continuous data between groups were performed using analysis of variance, followed by Student's $t$-tests.

**Data availability**. The authors declare that the data supporting the findings of this study are available within the paper and its supplementary information files.

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

## Acknowledgements

We thank Y.H. Jeon for supplying for materials (LRS protein and tRNA^Leu) and advice. This work was supported by the Global Frontier Project grant [NRF-2013M3A6A4072536] and [NRF-M3A6A4-2010-0029785] and by a grant from the Gyeonggi Research Development Program.

## Author contributions

J.H.K.: conception and design, collection and analysis of data, manuscript writing; C.L. and M.L.: conception and design, collection and analysis of data; H.W., K.K., S.J.P., I.Y., J.J., H.Z., H.K.K., N.H.K., S.J.J., H.C.Y., J.H.K., J.S.Y., M.Y.L., C.W.L., J.Y., S.J.O., J.S.K., and S.A.M.: collection and analysis of data; K.Y.H., M.G., and G.H.: conception and design; J.M.H.: conception and design, collection and analysis of data, manuscript writing, final approval of manuscript; S.K.: collection and analysis of data, manuscript writing, and final approval of manuscript.

# ARTICLE

## Additional information

**Competing interests:** The authors declare no competing financial interests.

