## [Peer Review File · Nature Communications]

Reviewers' Comments:

Reviewer #1 (Remarks to the Author)

I have been requested by the Editor to assess the structural modelling and SAXS data of the manuscript, therefore I will restrict my comments to this subject.

First at all, the SAXS data collection and analyse followed standard procedures and software allowing the authors to characterize at low resolution the conformational change arose upon BC-LI-0198 binding. Experimentally and methodologically the results are perfectly done. However, I do not understand the sentence "Comparison of the SAXS data between the naked and BC-LI-0198-bound LRS revealed no large conformational change resulting from the chemical binding (Rg and Dmax are 47.02Å and 205Å for the naked LRS, and 54.19Å and 230Å for the chemical-bound LRS). It is evident that both Rgs and SAXS envelopes have substantial variations. I can agree that the overall shape is similar and there is not large module rearrangement but it is clear that the bound structure is more elongate and thin indicating a conformational change.

I also have concerns about the modelling/fitting. Given the low resolution of the 3D models, the flexibility of the complex, and the relative low homology (32%) is not possible to perform a reliable and unique interpretation in terms of atomic models. In particular, the localization of at least, VC and UNE-L domains seems to be quite subjective. The authors justify this in pag 5 "Since the position of VC domain in the template structure was more extended due to tRNA binding³¹, we adjusted it closer to CD and ABD domain". Just by looking at the template structure it seems more reasonable that the VC domain will be even more extended rather than collapsed towards ABD domain. In fact, I submitted the LRS gene to the Xraptor modelling server (I attached here the model) and, at a glance, the VC/UNE-L model could fit better in the tip of your SAXS envelop. With this I do not saying that the X-raptor model is better than yours, I am illustrating that different interpretation can be made. In any case, in the fitting shown in figure 2b, the VC domain have been cut away from the C-term ABD and the UNE-L placed far away from the VC domain in both cases broken the logical sequence proximity. In principle, the authors should avoid the manual modification of the models to fit to the SAXS envelopes because is subjective and not reproducible. As I said before, the SAXS low resolution data, the accuracy of the modelling, the macromolecular flexibility of the systems prevents a reliable pseudo-atomic interpretations. Nevertheless, I think the fitting is an accessory element in the manuscript.

Minor points.

- Include standard errors in SAXS measurements (Rg & Dmax)
- Please, enlarge SAXS profiles in figure S2 B
- Please, use three orthogonal views to better display the atomic fit in the SAXS modelling. Given the low resolution of the data it is impossible to judge the quality of the fit from a single view.
 - The localization of potential missing regions (e.g. if modelling did not cover the 100% of structure) in the fitting will be valuable.
 - The authors should provide the sequence identity, coverage, modelling scores etc. per domain.

Reviewer #2 (Remarks to the Author)

In this manuscript, Kim et al. describe the development of a new inhibitor of the Leucyl-tRNA synthetase (LRS) which interferes with the binding of LRS with RagD GTPase and consequently blocks the mTOR pathway activation by leucine. This report is a logical and strong follow-up to the authors' previous publication on this subject in which they should that LARS functions as a GAP for RagD and thus controls mTOR activity. Development of a specific inhibitor for this interaction which at the same time does not abrogate the aminoacyl tRNA synthetase activity of the enzyme and thus limits the potential side-effects is a very important and promising therapeutic target for several types of cancers in which over-activation of mTOR is a driving factor.

The authors have used several approaches to investigate the specificity and the potency of their inhibitor in suppressing mTOR activity and cell growth/apoptosis of cancer cell lines as well as an in vivo tumor model. Therefore, this manuscript is a worthy of publication in Nature Communications. However, there are several aspects of the manuscript that need to be improved/strengthened as follows:

Major comments:

1. The authors showed that BC-LI-0186 does not bind or interfere with ARF2 or Sestrin2. However, does not exclude the possibility of binding or interfering with the activity of other proteins. Also, is the mitochondrial LRS homolog, LRS2, affected by this compound? Can the authors show that no other proteins in the cell are susceptible to this compound, at least not to the same extent as LRS?

It is not clear why the authors used different concentration ranges of the compound when testing LRS (Fig. 1E) and Sestrin2 (Fig. S1G). Would the results be the same if similar concentration ranges were used?

2. The authors conducted most of the initial experiments in measuring the specificity and efficiency of the compound with the BC-LI-0186, but the structural studies were performed with the more soluble BC-LI-0198. It is unclear if BC-LI-0198 also exhibit similar functional properties as BC-LI-0186.

3. The presentation style of Fig. 4E is quite confusing and perhaps misleading. How can there be more than 100% cell death in an experiment? The authors need to show cell death as a percentage of total cell number. Either way, it doesn't seem that the compound had a very huge effect on cell death. Also, what is the effect of the compound on the growth/death rate of normal cells? Could the authors test the compound on multiple cell lines for comparison? It is also noteworthy that the authors use different cell lines of different tumor types throughout the manuscript without proper justification on why each cell line was chosen for the indicated assay. Would the results be different with another cell type or a non-cancerous cell?

4. Fig. 7F and Supplementary Table 4; the authors described using BC-LI-0186 and rapamycin for blocking the in vivo tumor model. However, there is no clear explanation as to why the indicated concentration of BC-LI-0186 was used. This concentration is 20x more than rapamycin. Also, it is not clear how this could be a relevant comparison. Did the authors also perform this experiment with a range of concentrations of BC-LI-0186?

5. Fig. S1B; the authors examined the effect of BC-LI-0186 on several kinases. Interestingly the activity of GCN2 kinase was conspicuously increased, but there is no mention of the reason or potential impact of this effect on the results. Wouldn't activation of GCN2 which is usually a result of amino acid starvation, complicate the interpretation of the rest of the results?

Minor comments:

1. There are grammatical errors and confusing language throughout the text. It might help if the authors use an English editing service to improve the text. In addition, there are several errors that should be corrected. For instance, 'mTOCR1' to 'mTORC1' in page 1, 'Fig. 1b-c' to 'Fig. 3b-c' in page 6, 'S597A' to 'S974A' in page 8. SESN2 and sestrin2 are used interchangeably throughout the text.

2. The following sentence in page 7 is very confusing and does not match with the relevant figure. Does pS6K change upon treatment with the compound?

"To understand the working mechanism of the compound, we tested BC-LI-0186 whether it would affect the interaction between endogenous LRS and RagD and S6K phosphorylation, and observed its effective inhibition of the LRS-RagD interaction without any effect on LRS-Vps34, or LRS-EPRS binding, and the decreased S6K phosphorylation, respectively (Fig. 3e)."

3. Are the blots in Fig. 4B (lower panel) from the same gel? pS6K bands show different migrations/orientations than S6K.

4. In Fig. 4C, treatment of the cells transfected with WT LRS (+Leu) with BC-LI-0186 did not affect the S6K phosphorylation compared with the no-drug control. Shouldn't the BC-LI-0186-treatment suppress S6K phosphorylation under this condition?

5. In Fig. 5E, except in the first and last lanes, there seems to be no direct correlation between Parp and Cleaved-Parp bands. Can the authors explain this anomaly?

Reviewers' comments:

Reviewer #1

Q1. First at all, the SAXS data collection and analyse followed standard procedures and software allowing the authors to characterize at low resolution the conformational change arose upon BC-LI-0198 binding. Experimentally and methodologically the results are perfectly done. However, I do not understand the sentence “Comparison of the SAXS data between the naked and BC-LI-0198-bound LRS revealed no large conformational change resulting from the chemical binding (R_g and D_{max} are 47.02Å and 205Å for the naked LRS, and 54.19Å and 230Å for the chemical-bound LRS). It is evident that both R_g s and SAXS envelopes have substantial variations. I can agree that the overall shape is similar and there is not large module rearrangement but it is clear that the bound structure is more elongate and thin indicating a conformational change.

Answer: The reviewer made a good point. We agree with the concern about the phrase “conformational change” upon inhibitor binding. We initially meant that our data indicate no major change on overall shape or domain arrangement, but with some local conformational variants which result in elongated and thin SAXS envelope upon the inhibitor binding (as indicated by the changed R_g and D_{max}). We have modified the description about this part in the revised manuscript on Page 5, line 24.

Q2. I also have concerns about the modelling/fitting. Given the low resolution of the 3D models, the flexibility of the complex, and the relative low homology (32%) is not possible to perform a reliable and unique interpretation in terms of atomic models.

Answer: The absence of structures with higher sequence homology (higher than 30% sequence identity), made it difficult for precise model prediction. Fortunately, in the crystal structures of LRSs from different species (*Pyrococcus horikoshii*, *Escherichia coli*, *Thermus thermophilus*, *Mycoplasma mobile*, *Agrobacterium tumefaciens*) in PDB database (1wz2, 3zgz, 2v0g, 3ziu, 5ah5), the four shared domains (CP1, CD, ABD and VC domains) adopt similar arrangement, suggesting that 3D domain arrangement in LRSs would be conserved. Since the domain arrangement in the model from Phyre2 is similar to that found in the crystal structures, we consider that the prediction by Phyre2 is reasonable for the low-resolution analysis by SAXS. In addition, concerned with the linear information by SAXS, we also have cross-checked the potential conformational changes of LRS upon chemical binding by

hydrogen-deuterium exchange mass spectrometry (HDX-MS). Our HDX results showed that the CP1, CD and ABD (mainly CP1) have increased HDX upon binding to BC-LI-0198 (Fig. 2b), supporting more flexible conformation induced by the chemical binding at the VC domain. We inserted two additional modeling structures in Supplementary Fig. 3a and 3b.

Q3. In particular, the localization of at least, VC and UNE-L domains seems to be quite subjective. The authors justify this in page 5 “Since the position of VC domain in the template structure was more extended due to tRNA binding³¹, we adjusted it closer to CD and ABD domain”. Just by looking at the template structure it seems more reasonable that the VC domain will be even more extended rather than collapsed towards ABD domain. In fact, I submitted the LRS gene to the RaptorX modelling server (I attached here the model) and, at a glance, the VC/UNE-L model could fit better in the tip of your SAXS envelop. With this I do not say that the RaptorX model is better than yours, I am illustrating that different interpretation can be made.

Answer: It is a great point. For our model building, the following reasons have been used.

1. In the crystal structures of LRSs from other species (*Pyrococcus horikoshii*, *Escherichia coli*, *Thermus thermophilus*, *Mycoplasma mobile*, *Agrobacterium tumefaciens*) in PDB database (1wz2, 3zgz, 2v0g, 3ziu, 5ah5), VC domains are always involved in tRNA binding. Other LRS pdbs without tRNA do not have visible VC domain (flexible and disordered), with only one exception. *Mycoplasma mobile* LRS (MmLRS) with a visible VC domain was reported without binding tRNA in 2013 (Li et al., 2013). When comparing with the LRS vs LRS-tRNA (*Escherichia coli*, 4aq7), its VC domain was found to become more extended upon tRNA binding (Li et al., 2013).

2. The Phyre2 prediction selected a complex structure of *Pyrococcus horikoshii* LRS and tRNA as a top template for the modeling of the human cytosolic LRS. (Because it has more complete structure, containing 964 of 967 total PhLRS residues). Therefore, the modeling result may present the conformation of LRS with tRNA bound than the tRNA-free LRS. In tRNA-free LRS, VC domain may be closer to CD and ABD domains, as observed in MmLeuRS (3ziu, Li et al., 2013).

3. Third, upon SAXS envelope fitting, we discovered that the envelope for LRS VC domain region was not enough to fit the whole domain in, especially for the envelope upon the inhibitor binding, indicating a relative movement of the VC domain (by about 70 Å transition and about 90-degree rotation).

4. Fourth, the HDX (hydrogen-deuterium exchange) data indicated that there are increased solvent exposure on CP1, CD, ABD domains but not on VC domain, suggesting the VC domain itself might be relatively detached from the main structure of LRS in solution.

5. Lastly, these results suggest the VC domain is a flexible domain, which may be located more freely in solution state than in crystalline state when a stable crystal packing is not present. For a clearer comparison, we inserted the original Phyre2 program-predicted LRS model without modification into Supplementary Fig. 3a. However, this model appears to be less fit to the resolved SAXS envelope as below.

As to the suggestion of the reviewer using RaptorX, we performed the model prediction by RaptorX modeling server for LRS. After comparison, the RaptorX model resembles the one from Phyre2 and Robetta in general (0.883 Å rmsd for the 988 residues of RaptorX model to the 988 residues of Phyre2 model). However, we observed one significant difference that is the relative orientation of UNE-L domain to the other four domains, which makes it difficult to fit the RaptorX model into the SAXS envelopes as shown below.

As to your suggestion on different interpretation of possible structures, we conducted to come up with alternative model structure. Since several loop regions exist in VC domain, the position of VC domain and the overall shape of LRS might be more flexible as seen in other LRS structures. In fact, the predicted raw model from the Phyre2, was not perfectly matched to the SAXS envelops, especially to that of the LRS-BC-LI-0198 complex. Therefore, this model was further adjusted by moving the VC domain below the ABD domain and was inserted as an alternative model into Supplementary Fig. 3b.

Although the relative positions of VC and UNE domains are different between the two models (Fig. 2a and Supplementary Fig. 3b), our conclusion on the compound binding and its inhibitory effect on the LRS-RagD interaction would be the same.

Q4. In any case, in the fitting shown in figure 2b, the VC domain have been cut away from the C-term ABD and the UNE-L placed far away from the VC domain in both cases broken the logical sequence proximity. In principle, the authors should avoid the manual modification of the models to fit to the SAXS envelopes because is subjective and not reproducible. As I said before, the SAXS low resolution data, the accuracy of the modelling, the macromolecular flexibility of the systems prevents a reliable pseudo-atomic interpretations. Nevertheless, I think the fitting is an accessory element in the manuscript.

Answer: We appreciate the reviewer's concern. The VC domain is an inherently flexible domain in LRS. It was reported to become more extended after tRNA binding (Li et al., 2013). The template used for the prediction is a complex structure of PhLRS and tRNA because it is structurally more complete than other available naked LRS structures. There are two *Pyrococcus horikoshii* LRS structures in the PDB database, 1WZ2 for the LRS-tRNA complex which was used as template by Phyre2 prediction, and 1WKB for apo LRS. 1WZ2 contains 964 out of total 967 PhLRS residues, missing the three N-terminal residues while 1WKB contains 807 residues, missing the three N-terminal residues and the VC domain. The predicted model thus more represents the conformation of LRS bound to tRNA. In tRNA-free LRS crystalline structure, the VC domain is closer (14 Å closer) to the CD and ABD domains. Given that there are many flexible loop regions (up to 17) in the VC and UNE-L domains (found in both of the Phyre2 and RaptorX models), the relative position of VC and UNE-L domains would be flexible. Considering this flexibility of the VC domain, the SAXS envelope fitting helped us to understand the state of LRS and the potential action mechanism of the LRS inhibitor. Based on the experimental SAXS envelope, UNE-L domain (in both of the apo and inhibitor-bound LRS) is expected to be located at the bottom tip of the envelop (coherently from these independent data collection and analysis, shown in Fig. 2b and the answer to Q3). An approximately 45-degree rotation was needed to fit the UNE-L domain from the program-modelled raw models (Supplementary Fig. 3a), to the experimental envelopes of LRS in solution (the answer to Q3).

Minor points.

Q1. Include standard errors in SAXS measurements (Rg & Dmax)

Answer: The Rg values are 47.02 ± 1.01 and 54.19 ± 2.74 for apo and BC-LI-0198-bound LRS, respectively. Dmax is basically the radius that is used to integrate the electron X-ray diffraction during SAXS data processing and is larger than the entire molecule while excluding neighbors. Dmax values were optimal inputs in the program GNOM according to PDDF curve, the fitting between experimental and theoretical scattering curves. Since SAXS experiment generates only the 'best' Dmax, there is no standard errors for Dmax. We included Rg values in Supplementary Figure 3c.

Q2. Please, enlarge SAXS profiles in figure S2B

Answer: We showed enlarged SAXS profiles in Supplementary Figure 3c.

Q3. Please, use three orthogonal views to better display the atomic fit in the SAXS modelling. Given the low resolution of the data it is impossible to judge the quality of the fit from a single view.

Answer: The optimized figures are inserted into Fig. 2b.

Q4. The localization of potential missing regions (e.g. if modelling did not cover the 100% of structure) in the fitting will be valuable.

Answer: We previously tested model building using Phyre2 and Rosetta (We also used RaptorX as suggested by the reviewer). A larger structural model containing residues 13-1061 was built by Phyre2. Rosetta built the model for the UNE-L domain containing residues 1062-1176. So the final modeling covered nearly 100% of LRS protein (1164 residues out of 1176).

Q5. The authors should provide the sequence identity, coverage, modelling scores etc. per domain.

Answer: We used the full sequence of LRS to Phyre2 for structural model prediction and 928 residues have been modeled with 100% confidence by the single highest scoring template. Since full-length environment (domain-domain orientation, domain-domain interface interaction, domain-domain electrostatic field, domain-linker orientation, etc) is

critical to determine overall structure of entire protein, it is generally more accurate to model the structure with full-length information. Here, as the reviewer suggested, we re-performed the structural model prediction for CD, ABD and VC domains, individually, by Phyre2 and listed the parameters as below. The predictions were based on the templates c1qu2A, d1ivsa2 and c1wzB, respectively, all with sequence identity of 21%. 463 (94% sequence coverage), 126 (89% sequence coverage) and 135 residues (79% sequence coverage) were modeled for the three domains with confidence values of 100%, 100% and 98.5%, respectively. This information was added to the 3-Dimensional Structure Modeling of Method section (page 21).

Reviewer #2

Major comments:

Q1. The authors showed that BC-LI-0186 does not bind or interfere with ARF2 or Sestrin2. However, does not exclude the possibility of binding or interfering with the activity of other proteins. Also, is the mitochondrial LRS homolog, LRS2, affected by this compound? Can the authors show that no other proteins in the cell are susceptible to this compound, at least not to the same extent as LRS?

Answer: To further provide the answer to reviewer#2's comments, we analyzed the effects of BC-LI-0186 and BC-LI-0198 on the interactions of LRS-RagD, LRS-Vps34, LRS-EPRS, RagB-RagD, mTOR with Raptor and mLST8 and observed the following results. First, these two compounds specifically suppressed the interaction between LRS and RagD, but not between LRS and Vps34 or LRS and EPRS in LRS immunoprecipitation (upper panel). Second, while these two compounds suppressed the LRS-RagD interaction, they did not affect the RagB-RagD interaction in RagD immunoprecipitation (middle panel). In more detail, when RagD was immunoprecipitated with anti-RagD antibody in the presence of the compounds, RagB was still co-immunoprecipitated, but LRS was not. Third, these two compounds had no effect on the core complex of mTORC1 (lower panel). When mTOR was immunoprecipitated with anti-mTOR antibody, the amount of co-precipitated mLST8 and Raptor was not changed by the compound treatment. Furthermore, although BC-LI-0186 compound did not bind to Sestrin2 (Supplementary Fig. 1g), we further checked whether this compound affected the interaction between Sestrin2 and WDR24, which is a component of GATOR2 complex. As below, BC-LI-0186 compound as well as LRS knockdown had no effect on Sestrin2-WDR24 binding.

All these results suggest the specific inhibitory activity of BC-LI-0186 to the LRS-RagD interaction. Further, following the reviewer #2's additional comment, we also analyzed the effect of BC-LI-0186 on mitochondrial LRS (LRS2). The catalytic activity of LRS2 was affected by Leu-AMS, but not by BC-LI-0186. This data was combined with previous result in Supplementary Fig. 1c.

Q2. It is not clear why the authors used different concentration ranges of the compound when testing LRS (Fig. 1E) and Sestrin2 (Fig. S1G). Would the results be the same if similar concentration ranges were used?

Answer: We repeated the SPR analysis of Sestrin2 at the same concentration range of BC-LI-0186 and replaced Supplementary Fig. 1g with this data. Again, we obtained the same result that BC-LI-0186 does not bind to Sestrin2.

Q3. The authors conducted most of the initial experiments in measuring the specificity and efficiency of the compound with the BC-LI-0186, but the structural studies were performed with the more soluble BC-LI-0198. It is unclear if BC-LI-0198 also exhibit similar functional properties as BC-LI-0186.

Answer: In Supplementary Fig.2b, BC-LI-0198 directly binds to LRS. This compound suppressed leucine-induced S6K phosphorylation (Supplementary Fig. 2c). Furthermore, when LRS was immunoprecipitated with anti-LRS antibody, the amount of co-precipitated RagD was decreased by BC-LI-0198 as well as BC-LI-0186 (Fig. 3d, upper). These two compounds specifically suppressed LRS-RagD binding, but not RagB-RagD heterodimer formation (Fig. 3d, middle) and mTORC1 core complex formation (Fig. 3d, lower). In *in vitro* GTP-agarose pulldown assay, two compounds suppressed leucine-induced GTP hydrolysis of RagD (Fig. 4b). Also, the lysosomal localization of LRS and Raptor was increased by the treatment of leucine but not in the presence of BC-LI-0198 or BC-LI-0186 (Supplementary Fig. 4c). All of these results support that BC-LI-0198 would exhibit the same functional properties as BC-LI-0186.

Q4. The presentation style of Fig. 4E is quite confusing and perhaps misleading. How can there be more than 100% cell death in an experiment? The authors need to show cell death as a percentage of total cell number. Either way, it does not seem that the compound had a very huge effect on cell death. Also, what is the effect of the compound on the growth/death rate of normal cells? Could the authors test the compound on multiple cell lines for

comparison? It is also noteworthy that the authors use different cell lines of different tumor types throughout the manuscript without proper justification on why each cell line was chosen for the indicated assay. Would the results be different with another cell type or a non-cancerous cell?

Answer: In Supplementary Methods, we already described the detail of cell growth and death assays. Briefly, we used SW620 cell line stably expressing red fluorescence to monitor cell growth since the sum of red fluorescence is proportional to the number of cells. Thus, we think that we already showed the effect of BC-LI-0186 on total cell number (Fig. 4d). In Fig. 4e, we monitored the cell death using CellTox green dye and the basal level of control cells was calculated as 100%. To reduce conflict and misunderstanding, we changed the label of y-axis as Cytotoxicity Fluorescence (%). We determined the effect of BC-LI-0186 on the growth and death of different types of cancer cells and found that the compound severely suppressed the growth of colon and lung cancer cells, and less effectively pancreas, breast, ovarian, and glioblastoma cancer cells. We added this data into Supplementary Fig. 6a. Furthermore, in four different types of normal cells, human colon epithelial cells (FHC), human diploid lung fibroblast (WI-26), mouse embryo fibroblast (NIH/3T3), and human foreskin fibroblast (BJ), BC-LI-0186 (0.1, 5, 10, 20, 50, 100 μ M) and rapamycin (100 nM) had negligible effects on cell growth and death, whereas 5-FU (10 μ M) significantly affected both. We added these data into Supplementary Fig. 6b and 6c, respectively.

Q5. Fig. 7F and Supplementary Table 4; the authors described using BC-LI-0186 and rapamycin for blocking the *in vivo* tumor model. However, there is no clear explanation as to why the indicated concentration of BC-LI-0186 was used. This concentration is 20x more than rapamycin. Also, it is not clear how this could be a relevant comparison. Did the authors also perform this experiment with a range of concentrations of BC-LI-0186?

Answer: As a preliminary experiment, in HCT116 MW xenograft and HCT116 MM xenograft, we compared the efficacy of rapamycin at four different doses (0.5, 1, 2, and 4mpk). At 0.5 mpk of rapamycin, we couldn't see the suppressive activity in both xenografts. At 2 and 4 mpk of rapamycin, we observed the similar anti-tumor activities, suggesting that HCT116 MM cells were not rapamycin-resistant anymore in this *in vivo* condition. At 1mpk of rapamycin, we could see the meaningful difference of anti-tumor activity by showing that HCT116 MW and MM were rapamycin-sensitive and rapamycin-resistant, respectively, in *in vivo* xenograft model. Based on these results, we determined the working dose of rapamycin at 1mpk for *in vivo* experiment. Since BC-LI-0186 shows low solubility in PBS buffer and 3DW (Supplementary Table 1), "20mpk" was the maximal dose that could be used for the

experiments. The purpose of this experiment is to prove the anti-tumor activity of BC-LI-0186 in rapamycin-resistant tumor model *in vivo*, but not to compare the anti-tumor activity between BC-LI-0186 and rapamycin.

Q6. Fig. S1B; the authors examined the effect of BC-LI-0186 on several kinases. Interestingly the activity of GCN2 kinase was conspicuously increased, but there is no mention of the reason or potential impact of this effect on the results. Wouldn't activation of GCN2 which is usually a result of amino acid starvation, complicate the interpretation of the rest of the results?

Answer: Although *in vitro* GCN2 kinase activity looked somewhat enhanced by BC-LI-0186 (Supplementary Fig. 1b) as reviewer#2 mentioned, it did not induce eIF2 α S51 phosphorylation or GCN2 T899 auto-phosphorylation, which are markers of GCN2 activation, in cells (Supplementary Fig. 1f). Thus, we do not think that the slight increase of GCN2 kinase activity by the treatment of the compound is meaningful.

Minor comments:

Q1. There are grammatical errors and confusing language throughout the text. It might help if the authors use an English editing service to improve the text. In addition, there are several errors that should be corrected. For instance, 'mTOCR1' to 'mTORC1' in page 1, 'Fig. 1b-c' to 'Fig. 3b-c' in page 6, 'S597A' to 'S974A' in page 8. SESN2 and sestrin2 are used interchangeably throughout the text.

Answer: We corrected all errors that reviewer #2 pointed out.

Q2. The following sentence in page 7 is very confusing and does not match with the relevant figure. Does pS6K change upon treatment with the compound? "To understand the working mechanism of the compound, we tested BC-LI-0186 whether it would affect the interaction between endogenous LRS and RagD and S6K phosphorylation, and observed its effective inhibition of the LRS-RagD interaction without any effect on LRS-Vps34, or LRS-EPRS binding, and the decreased S6K phosphorylation, respectively (Fig. 3e)."

Answer: We modified this sentence more clearly. "Next, we examined the specificity of BC-LI-0186 for its interaction with LRS and RagD. BC-LI-0186 and its derivative BC-LI-0198 specifically inhibited the interaction of LRS and RagD, but had little effect on interactions of LRS-Vps34³², LRS-EPRS³³, RagB-RagD pairs⁷, and the core complex formation of

mTORC1³⁴ (Fig. 3d).”

Q3. Are the blots in Fig. 4B (lower panel) from the same gel? pS6K bands show different migrations/orientations than S6K.

Answer: In Fig. 4b of previous manuscript, we got the western blot bands individually using the same lysates of each sample. As reviewer #2 suggested, we repeated this experiment and got the same result from the same gel. We replaced Fig. 4b with this new data.

Q4. In Fig. 4C, treatment of the cells transfected with WT LRS (+Leu) with BC-LI-0186 did not affect the S6K phosphorylation compared with the no-drug control. Shouldn't the BC-LI-0186-treatment suppress S6K phosphorylation under this condition?

Answer: Since BC-LI-0186 directly binds to LRS (Fig. 1e), overexpressed LRS may compensate for the inhibitory activity of BC-LI-0186. To confirm that LRS is a major effective target for BC-LI-0186, we examined the inhibitory effect of BC-LI-0186 on mTORC1 activity in SW620 cells expressing DOX-inducible LRS WT or S974A. In previous Fig. 4c, we observed that the BC-LI-0186-dependent inhibition of S6K phosphorylation was recovered when LRS expression was induced by 1 μ g/ml DOX treatment. As below, LRS expression was saturated by 1 μ g/ml DOX treatment.

To compare the compensatory potency of LRS WT or S974A on BC-LI-0186, we examined the dose-dependent effect of DOX (0, 0.25, 1 μ g/ml) on BC-LI-0186-suppressed S6K phosphorylation. Interestingly, at 0.25 μ g/ml of DOX, overexpressed LRS S974A completely rescued the level of phospho-S6K while overexpressed WT partially rescued it. At 1 μ g/ml of DOX, overexpression of WT or S974A completely rescued it (Fig. 4c). Thus, this data also supports that LRS is a specific target for BC-LI-0186.

Q5. In Fig. 5E, except in the first and last lanes, there seems to be no direct correlation between Parp and Cleaved-Parp bands. Can the authors explain this anomaly?

Answer: In this study, we used anti-PARP antibody (Cell Signaling, #9542), which is a rabbit polyclonal antibody, to detect endogenous full length PARP1 as well as the cleaved fragment of PARP1. From the web site of Cell Signaling Technology, we can see the western blot result with this antibody. In Jurkat cells, the cleaved PARP1 was detected by etoposide treatment without any dramatic change of full length PARP1.

Western blot analysis of extracts from NIH/3T3 cells, untreated or staurosporine-treated (1 μ M), and Jurkat cells, untreated or etoposide-treated (25 μ M), using PARP Antibody.

From the website of Cell Signaling Technology

From the figures of two different references (Nature 405:974-978 (2000); Clinical Cancer Research, 18(20):5639-5649 (2012)), we can find the same blot pattern. Thus, we think that this result should be understood qualitatively.

Fig.3c of Nature 405:974-978 (2000)

Fig. 1A of Clinical Cancer Research, 18(20):5639-5649 (2012)

Reviewers' Comments:

Reviewer #1:

Remarks to the Author:

I insist, given the low resolution of the 3D models, the flexibility of the complex, and the relative low homology is not possible to perform a reliable and unique interpretation in terms of atomic models. You have at least three important sources of error 1) The limited resolution of the SAXS models that prevents an unambiguous fitting 2) The homology models with <30% of identity are likely to contain errors 3) The flexibility of your system. The future readers of your article should be aware of the limitations of your modeling. Please, add a couple of sentences accordingly.

To me, your untouched model (Figure 3a in your rebuttal letter) have the same predictive value, if not better (it is reproducible), than the model of figure 2. In fact, I could argue with you that the LRS fitting of the original model, is as good as your final manually model. For example, CD and ABD domains seems to be too close to the surface, and there are a few residues sticking out the density envelop. The density not covered for the model can be explained by the lack of resolution, flexibility and modeling errors. The fact that the VC domain does not fit in the LRS+BC-LI-0198 is not a problem. You have convincing arguments to stand that VC domain is highly flexible (points 1-4 of your answer to Q3). If you want to extend the modelling (I do not think is mandatory), a more elegant way will be manually flex this domain but maintaining the integrity of the chain (you can do this with Maestro for example). Cutting and paste domains and moving them goes against the integrity/quality of your initial model. Correct if I am wrong, but your models of Figure 2, 3a, 3b are all compatible with your experimental data, as you said, "conclusion on the compound binding and its inhibitory effect on the LRS-RagD interaction would be the same". Also all models, knowing the flexibility of VC and the limited resolution of the SAXS data, are also compatible in terms of fitting. In this context, a raw modeling is a more safe option to prevent "overfitting" interpretations. Alternatively, you should include all the manually curated fitting models as supplementary material, at least to show that multiple interpretations are possible. Also, include the corresponding fitting figures of your rebuttal letter in supplementary material.

Q1. I am glad that you realize that you have a conformational change. The term "local conformational change" is misleading. In Structural Biology local change is often related to a very localized change, e.g. a conformational change of a binding loop. Your SAXS data support a substantial conformational change, the Rg change from 47.02 to 54.19 can be only explained by a domain (or domains) rearrangement. At least, in page 6 remove the term "local".

Q2. 1.) If you are sure about 4aq7 have a more realistic localization of VC domain you should use it as a template in the modeling. Please, include arguments 1-4 more clearly in the main text.

Q4. Sorry but I still have concerns about manually cut and move domains disregarding the maintenance of the covalent structure of your model. The resolution of your SAXS envelop is VERY low you CAN NOT fit unambiguous a single domain. Sentences like "An approximately 45-degree rotation was needed to fit the UNE-L domain from the program-modelled raw models" is an overinterpretation of your data. In fact, at a glance, the UNE-L location of your fitted model without modification (figure S3a, letter) is as good as your 45 degree manually rotated (figure 2). Correct me if I am wrong, it seems that you make a 45 rotation in the LRS case but not in the bound state (comparing figure 2 with 3a/3b of your letter). What does it mean? The binding of BC-LI-0918 recovers the conserved orientation extracted in the homology modeling that the apo LRS lacks. Do you have additional data that support this?

Also, I have no doubts that this system is flexible, but if you give some credit to your homology model you cannot break it apart ignoring the contiguity of the chain. When you move away the domains, are the distance between Ct- and Nt terminals of the fragments compatible with your initial model?

Minor.

If you are covering the 100%, (Q2) you should comment why there are extra densities not accounted by the model (again poor resolution, flexibility, errors in the modeling etc.).

Please check caption of b of S3

There are two beta sheets of CP1 in the tip your original model (fig 3a) that I do not see in the "manual" fitted model. Are they removed?

Reviewer #2:

Remarks to the Author:

The authors have satisfactorily addressed most of my comments. I have a few additional suggestions for the authors.

1. The authors already showed the leucine-specific effect of BC-LI-0186 on the inhibition of S6K phosphorylation as compared to glutamine and arginine in Supplementary Fig. 1a. How about the inhibitory effect of BC-LI-0186 on the S6K phosphorylation mediated by isoleucine, which also induces S6K phosphorylation via LRS shown in the authors' previous work? In addition, is BC-LI-0186 able to inhibit amino acid-induced 4E-BP1 phosphorylation? The authors can examine using a simple experiment as shown in Supplementary Fig. 1a.
2. Can the authors improve the images shown in Fig 5b, Supplementary Fig. 5a, and 5c? It is hard to correlate the graph shown in Supplementary Fig. 5e with the images shown in Supplementary Fig 5c. The authors must state the nucleotide binding status of RagB/RagD when they were used as mutated forms to prevent misleading. For example, in Supplementary Fig. 5c-e, does 'RagB/D' means 'RagBGTP/RagDGDP'?
3. Fig. 3f; please add 'Myc-LRS' for EV, WT, S974A, and S953A.

Answers to Comments

Reviewer #1:

I insist, given the low resolution of the 3D models, the flexibility of the complex, and the relative low homology is not possible to perform a reliable and unique interpretation in terms of atomic models. You have at least three important sources of error 1) The limited resolution of the SAXS models that prevents an unambiguous fitting 2) The homology models with <30% of identity are likely to contain errors 3) The flexibility of your system. The future readers of your article should be aware of the limitations of your modeling. Please, add a couple of sentences accordingly. To me, your untouched model (Figure 3a in your rebuttal letter) have the same predictive value, if not better (it is reproducible), than the model of figure 2. In fact, I could argue with you that the LRS fitting of the original model, is as good as your final manually model. For example, CD and ABD domains seems to be too close to the surface, and there are a few residues sticking out the density envelop. The density not covered for the model can be explained by the lack of resolution, flexibility and modeling errors. The fact that the VC domain does not fit in the LRS+BC-LI-0198 is not a problem. You have convincing arguments to stand that VC domain is highly flexible (points 1-4 of your answer to Q3). If you want to extend the modelling (I do not think is mandatory), a more elegant way will be manually flex this domain but maintaining the integrity of the chain (you can do this with Maestro for example). Cutting and paste domains and moving them goes against the integrity/quality of your initial model. Correct if I am wrong, but your models of Figure 2, 3a, 3b are all compatible with your experimental data, as you said, “conclusion on the compound binding and its inhibitory effect on the LRS-RagD interaction would be the same”. Also all models, knowing the flexibility of VC and the limited resolution of the SAXS data, are also compatible in terms of fitting. In this context, a raw modeling is a more safe option to prevent “overfitting” interpretations. Alternatively, you should include all the manually curated fitting models as supp. material, at least to show that multiple interpretations are possible. Also, include the corresponding fitting figures of your rebuttal letter in supplementary material.

Answer: We agree with reviewer#1’s kind recommendation. In a revised manuscript, we only included the predicted structural model and excluded the manual fitting model to

prevent “overfitting” interpretations. Thanks for valuable advice that make our result more consistent and concrete.

Q1. I am glad that you realize that you have a conformational change. The term “local conformational change” is misleading. In Structural Biology local change is often related to a very localized change, e.g. a conformational change of in binding loop. Your SAXS data support a substantial conformational change, the R_g change from 47.02 to 54.19 can be only be explained by a domain (or domains) rearrangement. At least, in page 6 remove the term “local”.

Answer: As reviewer #1 recommended, we removed the term “local” in page 6.

Q2. If you are sure about 4aq7 have a more realistic localization of VC domain you should use it as a template in the modeling. Please, include arguments 1-4 more clearly in the main text.

Answer: The *PhLRS* was chosen as a model for *hsLRS* based on higher sequence homology. Since the unmodified model is only shown in the structural analysis, relocation of VC domain is no longer considered in the revised manuscript.

Q3. Sorry but I still have concerns about manually cut and move domains disregarding the maintenance of the covalent structure of your model. The resolution of your SAXS envelop is VERY low you CAN NOT fit unambiguous a single domain. Sentences like “An approximately 45-degree rotation was needed to fit the UNE-L domain from the program-modelled raw models” is an overinterpretation of your data. In fact, at a glance, the UNE-L location of your fitted model without modification (figure S3a, letter) is as good as your 45 degree manually rotated (figure 2). Correct me if I am wrong, it seems that you make a 45 rotation in the LRS case but not in the bound state (comparing figure 2 with 3a/3b of your letter). What does it mean? The binding of BC-LI-0918 recovers the conserved orientation extracted in the homology modeling that the apo LRS lacks. Do you have additional data that support this? Also, I have not doubts that this system is flexible, but if you give some credit to your homology model you cannot break it apart ignoring the contiguity of the chain. When you move away the domains, are the distance between Ct- and Nt terminals of the fragments

compatible with your initial model?

Answer: As reviewer #1 recommended, we modified the text and figures, omitting the manual fitting model for consistency and simplicity. We appreciate for valuable advice that makes our result more consistent and concrete.

Minor.

If you are covering the 100%, (Q2) you should comment why there are extra densities not accounted by the model (again poor resolution, flexibility, errors in the modeling etc.).

Answer: We added the sentence “The extra SAXS densities beyond the *hs*LRS structure may be from lower resolution of the SAXS, and potentially, suggest the flexibility of VC domain in LRS structure” in the line 15 of page 5.

Please check caption of b of S3

Answer: We corrected the caption of Supplementary Fig. 3b.

There are two beta sheets of CP1 in the tip your original model (fig 3a) that I do not see in the “manual” fitted model. Are they removed?

Answer: As reviewer#1 suggested, we removed all manual fitting models for consistency and simplicity. In Fig. 2a and b, we replaced CP1 domain structure with real human LRS-CP1 domain (which has a structure in the pdb).

Reviewer #2:

The authors have satisfactorily addressed most of my comments. I have a few additional suggestions for the authors.

Q1. The authors already showed the leucine-specific effect of BC-LI-0186 on the inhibition of S6K phosphorylation as compared to glutamine and arginine in Supplementary Fig. 1a. How about the inhibitory effect of BC-LI-0186 on the S6K phosphorylation mediated by

isoleucine, which also induces S6K phosphorylation via LRS shown in the authors' previous work? In addition, is BC-LI-0186 able to inhibit amino acid-induced 4E-BP1 phosphorylation? The authors can examine using a simple experiment as shown in Supplementary Fig. 1a.

Answer: As reviewer #2 suggested, we repeated this experiment and replaced Supplementary Fig. 1a with new data. BC-LI-0186 is able to inhibit leucine- and isoleucine-induced phosphorylation of S6K and 4EBP1.

Q2. Can the authors improve the images shown in Fig 5b, Supplementary Fig. 5a, and 5c? It is hard to correlate the graph shown in Supplementary Fig. 5e with the images shown in Supplementary Fig 5c. The authors must state the nucleotide binding status of RagB/RagD when they were used as mutated forms to prevent misleading. For example, in Supplementary Fig. 5c-e, does 'RagB/D' means 'RagBGTP/RagDGDP'?

Answer: As reviewer #2 recommended, we repeated these experiments and replaced Fig. 5b, Supplementary Fig.5a and 5c with new data. In Supplementary Fig. 5d and e, we marked RagB^{GTP}/D^{GDP} for preventing misleading.

Q3. Fig. 3f; please add 'Myc-LRS' for EV, WT, S974A, and S953A.

Answer: We inserted "Myc-LRS" in Fig. 3f.

Reviewers' Comments:

Reviewer #1:

Remarks to the Author:

The authors successfully addressed all my previous concerns.

Reviewer #2:

Remarks to the Author:

I think the revised paper is greatly improved and will be of wide interest to the community.